# Spatial segregation of catalytic sites within Pd doped H-ZSM-5 for fatty acid hydrodeoxygenation to alkanes

Shengzhe Ding [1,2], Dario Luis Fernandez Ainaga[3], Min Hu[1], Boya Qiu[1], Ushna Khalid[1], Carmine D'Agostino [1,4], Xiaoxia Ou [1,5], Ben Spencer [6,7], Xiangli Zhong [6,7], Yani Peng[1], Nicole Hondow [3], Constantinos Theodoropoulos[1], Yilai Jiao[8], Christopher M. A. Parlett [1,9,10,11] ✉ & Xiaolei Fan [1,5,12] ✉

Spatial control over features within multifunctional catalysts can unlock efficient one-pot cascade reactions, which are themselves a pathway to aviation biofuels via hydrodeoxygenation. A synthesis strategy that encompasses spatial orthogonality, i.e., one in which different catalytic species are deposited exclusively within discrete locations of a support architecture, is one solution that permits control over potential interactions between different sites and the cascade process. Here, we report a Pd doped hierarchical zeolite, in which Pd nanoparticles are selectively deposited within the mesopores, while acidity is retained solely within the micropores of ZSM-5. This spatial segregation facilitates hydrodeoxygenation while suppressing undesirable decarboxylation and decarbonation, yielding significant enhancements in activity (30.6 vs 3.6 $mol_{dodecane} \, mol_{Pd}^{-1} \, h^{-1}$) and selectivity ($C_{12}$:$C_{11}$ 5.2 vs 1.9) relative to a conventionally prepared counterpart (via wet impregnation). Herein, multifunctional material design can realise efficient fatty acid hydrodeoxygenation, thus advancing the field and inspiring future developments in rationalised catalyst design.

Cascade reactions, particularly catalytic cascades, are powerful strategies for complex multistep chemical processes within pharmaceuticals[1], agrochemicals[2], and biorefining[3,4]. Such synthetic approaches can reduce the number of product isolation steps while also permitting alternative reaction pathways that may not otherwise be feasible, for example, where an unstable intermediate prevents its isolation[5,6]. This reduction in chemical separations increases the overall energy and atom efficiency, benefiting process economics[7]. One-pot catalytic cascades typically require coupling of at least two individual catalytic materials or combining two or more catalytic species on a single material, i.e. multifunctional materials[8–11]. The latter, in particular, is already showing great potential, including for reactions such as Fischer-Tropsch[12,13], transesterification[9,14] and xylose valorisation to γ-valerolactone[15,16]. These have necessitated the amalgamation of a range of different pairs

[1]Department of Chemical Engineering, The University of Manchester, Manchester M13 9PL, UK. [2]Institute of Catalysis Science, Beijing Research Institute of Chemical Industry, Sinopec, Beijing 100013, China. [3]School of Chemical and Process Engineering, University of Leeds, Leeds LS2 9JT, UK. [4]Dipartimento di Ingegneria Civile, Chimica, Università di Bologna, 40131 Bologna, Italy. [5]Nottingham Ningbo China Beacons of Excellence Research and Innovation Institute, Ningbo 315100, China. [6]Henry Royce Institute, The University of Manchester, Manchester M13 9PL, UK. [7]Department of Materials, The University of Manchester, Manchester M13 9PL, UK. [8]Shenyang National Laboratory for Materials Science, Chinese Academy of Sciences, Shenyang 110016, China. [9]Diamond Light Source, Harwell Science and Innovation Campus, Didcot, Oxfordshire OX11 0DE, UK. [10]University of Manchester at Harwell, Harwell Science and Innovation Campus, Didcot, Oxfordshire OX11 0DE, UK. [11]UK Catalysis Hub, Rutherford Appleton Laboratory, Harwell, Oxfordshire OX11 0FA, UK. [12]Institute of Wenzhou, Zhejiang University, Wenzhou 325006, China. ✉e-mail: christopher.parlett@manchester.ac.uk; xiaolei.fan@manchester.ac.uk

of active sites, including metal and acid, acid and base, and Lewis and Brønsted acids. The combined sites either act alone, with one site for each distinct step of the cascade[17], or synergistically to promote one step[18] or the entire cascade[19]. Furfural upgrading to γ-valerolactone, is an example of the former, with transfer hydrogenation of the carbonyl group over Lewis acid sites and subsequent hydrolytic ring-opening over Brønsted acid sites[20]. Conversely, phenol hydroalkylation requires the interaction of metal and acid sites[21,22]. The rational design of multifunctional catalysts for cascade processes, therefore, requires attention regarding the reaction sequence and the mechanisms of each step. Furthermore, the order in which the distinct catalytic species are encountered within the cascade process is also critical if there is the potential for side reactions or unwanted interactions[9]. Thus, consideration of whether a random distribution of sites is acceptable or if the cascade dictates that careful engineering of the spatial location of different active sites (i.e. segregated) is needed is also imperative.

A core-shell structure has shown to be beneficial towards Fischer-Tropsch synthesis, with intermediate olefins (>$C_5$), produced at a metallic core, further converted to aromatic species within the microporous zeolite shell. The micropores give rise to prolonged diffusion, which assists the final step of the process[11,23]. Alkane isomerisation and cracking can likewise be tuned from nanoscale active site separation either within a core-shell architecture or on a binder zeolite composite, with control over metal deposition sites of the latter based on the difference between isoelectric points of alumina and zeolites[24,25]. Spatially orthogonal catalysts based on a hierarchical porous network have also resulted from their nanoscale design, with the mesopores and macropores of a silica framework selectively functionalised with two different catalytic species to govern the catalytic reaction sequence[9,26]. However, basing the design on duel templated mesoporous macroporous silicas can require controlled selective extraction of the two different pore templating agents individually.

Biofuels, preferably from feedstocks that do not compete with food supplies, have the potential to, in part, address carbon emissions through the replacement of fossil fuels. The aviation sector is one critical sector where biofuels can play a key role, given the steady growth of global air traffic demand (~5% per year) and the lack of other fuel alternatives at appropriate technology maturity[27,28]. Hydrodeoxygenation (HDO) of fatty acids (and triglycerides) to alkanes, ideally under mild reaction conditions (<300 °C), provides one of the most promising economically viable solutions under investigation[29,30]. Much of the focus to date has encompassed the union of noble metal catalytic sites with strong acidity, the latter vital to driving the HDO process. For example, Pd/C and Pd/$Al_2O_3$ strongly favour decarboxylation/decarbonylation ($DCO_x$), releasing $CO_2$ and CO as by-products, when operating at high process temperatures (>300 °C)[29]. Pd/$SiO_2$ and Pd/$TiO_2$ are active for succinic acid HDO, although selectivity often favours (slightly) the cyclic γ-Butyrolactone, which can be envisaged as an intermediate of the diol, with Pd particle size a critical parameter[31,32]. Adding acidity to the catalytic system, either by employing alumina silicates or through grafting heteropolyacids, is beneficial, introducing a propensity for HDO while simultaneously promoting fatty acid conversion[33-35]. The reaction pathway, based on experimental and kinetic modelling of biomass-derived species, is considered a multistep cascade process. Carboxylic acids are sequentially reduced to the corresponding alcohol, via the aldehyde intermediate, over metal sites. Subsequent alcohol dehydration yields alkenes, which in turn can undergo facile hydrogenation to saturated hydrocarbons while retaining the original carbon chain length. The alcohol intermediate may also undergo direct hydrogenation to the alkane, attributed to the absence of alkene detection, although this may reflect the significant low activation energy of C=C hydrogenation, while the formation of esters, ethers and aldol condensation products are other potential side reactions[36,37]. These latter pathways provide a circular route via their conversion back into the constituent species. Decreased carbon chain

length species result from decarboxylation and decarbonylation, with the intermediate aldehyde a potential candidate for the latter, for which Ni/ZSM-5 has shown high and tunable selectivity towards[37,38]. Overall, catalyst composition governs the propensity towards this array of different reactions (e.g. HDO, $DCO_x$, coupling), with choice of metal, acidity (including that of the support) and mass transfer, either singularly or combined, controlling factors[37,38]. Bimetallic systems, specifically incorporating Lewis acid sites (e.g. $MoO_x$, $Nb_2O_5$, $ReO_x$) into Pd or Pt based catalysts, also show potential for HDO through switching off the $DCO_x$ pathway[39,40]. Nevertheless, in the case of Re, an optimal promoter, the synthesis condition can be critical due to the multi-oxidation state species reported to be responsible, the necessity for the two metals to be adjoined[41], and the potential for Re leaching from the final catalyst[42]. Furthermore, Brønsted acidity may need to be added, achievable through a physical mixture with a zeolite, if alkane production is to be realised[43], while low operating temperatures of the bimetallic system, typically less than 200 °C, can take a toll on activity.

Here, we report the development of a simple strategy to prepare multifunctional catalytic materials with spatially segregated metal and acidic sites based on industrially relevant zeolite materials. Hierarchical mesoporous-microporous ZSM-5 zeolite is deployed as both catalysts, given its inherent acidity, and catalyst support to host deposited metal sites. The combination of Brønsted acid sites and Pd has shown the capacity for HDO, but to date, not within spatially orthogonal systems. Active site location is governed by size exclusion effects, with acid sites selectively extracted from the mesopores and replaced with preformed Pd metal nanoparticles (NPs). The choice of a mono-metallic Pd species over a bimetallic system was to entrench a degree of simplicity within what is already a multistep synthesis, albeit one amenable to scale-up. The spatial segregation of these two distinct catalytic active species shows profound benefit for the one-pot cascade HDO of the lauric acid to $C_{12}$ alkanes, a model reaction for catalytic HDO fatty acid upgrading to biofuel equivalents of diesel and kerosene. This design strategy represents a generic platform for multifunctional materials, including catalysts of industrial interest, in particular for producing those in which spatial separation of different chemical functionalities is desired.

## Results and discussion
### Controlling Pd site location within hierarchical ZSM-5

Hierarchical H+ form ZSM-5 (H-MZSM5) arises from the post-synthetic treatment of the parent zeolite (H-ZSM-5) with NaOH, evidenced by a characteristic Type IV isotherm, Barrett–Joyner–Halenda (BJH) pore size analysis, and electron microscopy, as shown in Fig. 1 and Supplementary Fig. 1. The formation of mesopores, with an average diameter of 8.6 nm (Fig. 1b), gives rise to a fivefold increase in mesopore volume, while a relative crystallinity of 80.6% and comparable micropore volume confirm preservation of micropore network (Supplementary Table 1 and Supplementary Fig. 1b). The presence of mesopores is subsequently exploited for the selective deposition of metal NPs solely within these domains through size exclusion, as illustrated in Fig. 1a. Pd NPs, possessing an average diameter of 4.6 ± 1.3 nm (Supplementary Fig. 2), are deposited exclusively outside of the micropores of the hierarchical architecture (PdNP/H-MZSM5). This spatial control is governed by the difference in pore diameter, 8.6 nm (meso) vs 0.54−0.56 nm (micro), thus, preformed Pd NPs can reside only at the external surface and/or within the mesopores of H-MZSM5. Furthermore, spatial separation of Pd and acid sites is realised with Brønsted acid sites (i.e. Al-O(H)-Si) principally located within the retained micropores. Conversely, conventional impregnation and thermal processing of a Pd salt (Pd$_{imp}$/H-MZSM5) cannot achieve this active site spatial separation (Fig. 1).

High-angle annular dark-field scanning transmission electron microscopy (HAADF-STEM) of PdNP/H-MZSM5 and Pd$_{imp}$/H-MZSM5 shows a distinct difference in the distribution and size of Pd within the

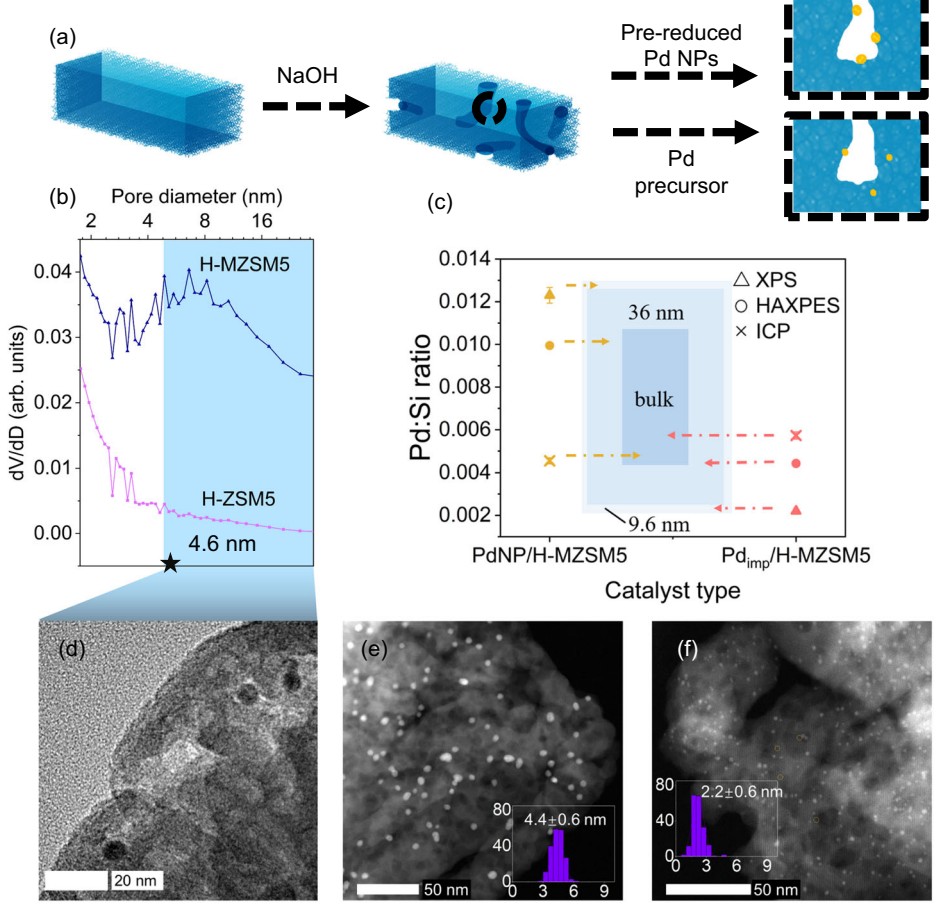

**Fig. 1 | Catalyst synthesis and physicochemical properties. a** Graphic illustration of catalyst synthesis. **b** Pore size distribution based on the BJH method with Faas correction using the adsorption branch of $N_2$ physisorption isotherms. **c** Pd:Si molar ratio by X-ray photoelectron spectroscopy (XPS), hard X-ray photoelectron spectroscopy (HAXPES) and inductively coupled plasma optical emission spectroscopy (ICP). **d** TEM image of PdNP/H-MZSM5. **e** HAADF-STEM image of PdNP/H-MZSM5 (inset: Pd particle size distribution). **f** HAADF-STEM image of Pd$_{imp}$/H-MZSM5 (inset: Pd particle size distribution). The error on the average particle sizes reported is the standard deviation of the dataset.

porous support framework. The conventional preparation route, namely Pd$_{imp}$/H-MZSM5, produces highly dispersed small Pd NPs (2.2 ± 0.6 nm) in conjunction with sub-1-nm Pd clusters, as shown in Fig. 1f and Supplementary Fig. 1. These species are randomly distributed throughout the three-dimensional hierarchical ZSM-5 skeleton, including within micropores, as confirmed by HAADF-STEM energy dispersive x-ray spectroscopy (EDX), consistent with previous investigations in which metal nanoparticles larger than the micropore diameter have been confirmed inside the zeolite framework[44]. Thus, such a synthesis approach endorses Pd integration within the micro and mesopores, i.e. uncontrollably across the entire hierarchical pore network and hence non-selectively[45]. The combination of Pd NPs and clusters results in a Pd dispersion of 26.5% (Supplementary Table 2). In contrast, PdNP/H-MZSM5 displays only Pd NPs, with an average diameter of 4.4 ± 0.6 nm (Fig. 1e), which are consistent with the parent unsupported NPs and naturally possess a lower dispersion (11.9%, Supplementary Table 2). Therefore, given their sizes and the size of the two distinct pore domains, and in conjunction with the imaging, Pd NPs reside selectively and exclusively in the mesopores and outer surface of the support matrix, as evidenced by Fig. 1d, e, for PdNP/H-MZSM5, whilst Pd is present in the micro and mesopores in Pd$_{imp}$/H-MZSM5 (Fig. 1f). It is worth pointing out that incorporating Pd, via either route, does not lead to micropore blockage. Comparable micropore properties to the parent zeolite (Supplementary Table 1)

provides evidence of the highly accessible hierarchical framework structure.

Further evidence of the difference in Pd distribution within the two catalysts is afforded through the combination of inductively coupled plasma (ICP) (Supplementary Table 2), X-ray photoelectron spectroscopy (XPS) and hard X-ray photoelectron spectroscopy (HAXPES) (Fig. 1c and Supplementary Fig. 3). Bulk elemental analysis (ICP) confirms comparable Pd:Si ratios, 0.0045 for 0.81 wt% PdNP/H-MZSM5 and 0.0057 for 1.02 wt% Pd$_{imp}$/H-MZSM5, respectively. Due to the inherent sampling depths of XPS and HAXPES, ~9.6 nm for Al K$\alpha$ vs ~36 nm for Ga K$\alpha$, for which 63% of the signal originates from the top 3.2 and 12 nm, respectively, they provide pseudo surface vs pseudo bulk analysis[46], i.e. they probe depth-dependence of Pd species across the two catalysts. A significantly higher Pd concentration at the surface (external and within directly accessible mesopores) is identified by XPS for PdNP/H-MZSM-5, whilst HAXPES reveals an intermediate Pd:Si ratio of the three techniques. This trend completely flips for Pd$_{imp}$/H-MZSM5, with comparable ratios for HAXPES and ICP and a significant drop for XPS, indicative of significant levels of Pd migration into the bulk of the zeolite skeleton during impregnation, a feat not observed of PdNP/H-MZSM5. Both catalytic materials consist of Pd metal, with only a trace amount of PdO detected in Pd$_{imp}$/H-MZSM5, again reflecting the smaller NP size and, thus, increase in surface-to-bulk ratio and, therefore, greater surface PdO concentration[47].

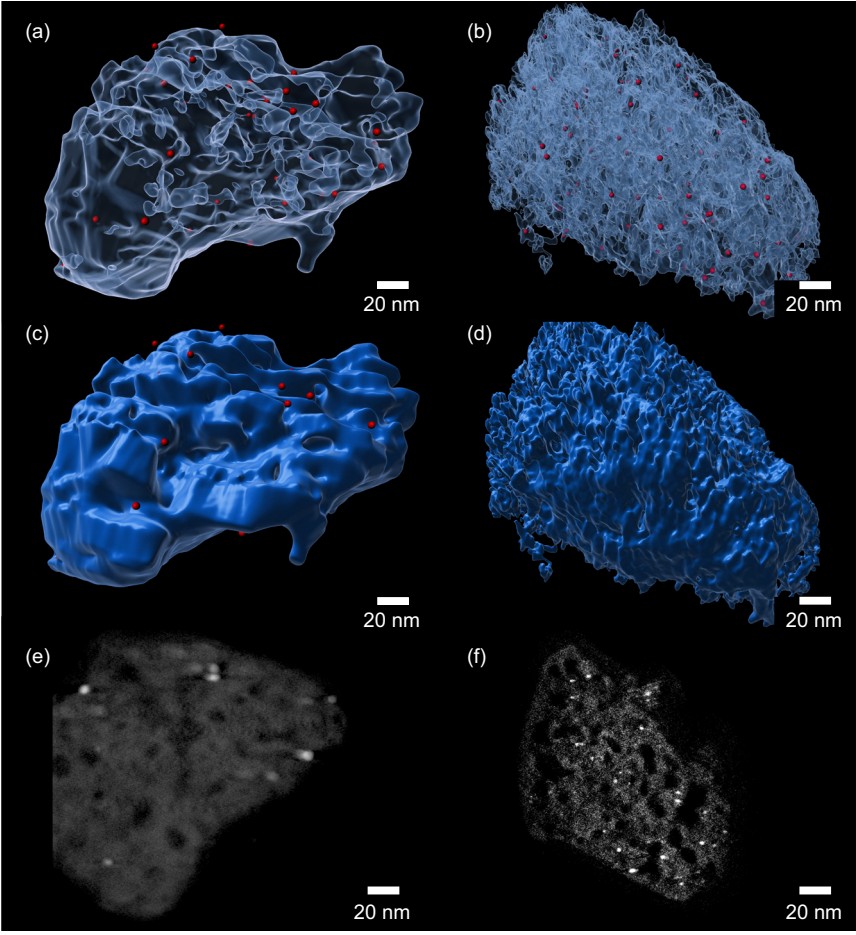

**Fig. 2 | STEM tomogram reconstructions. a, c** PdNP/H-MZSM5 and **b, d** Pd$_{imp}$/H-MZSM5, with the Zeolite set as semi-transparent in (**a, b**) and non-transparent in (**c, d**). Cross-section slices from the centre of the catalyst are presented in **e** PdNP/H-MZSM5 and **f** Pd$_{imp}$/H-MZSM5.

Three-dimensional tomography reconstructions of PdNP/H-MZSM5 and Pd$_{imp}$/H-MZSM5, produced from tilt-series HAADF-STEM imaging, were established to study the distribution of Pd NPs through the two catalysts (Fig. 2 and Supplementary Videos 1–4) and further validate the distinct difference in Pd location between the two catalysts. Semi and non-transparent rendering of the zeolite framework, Fig. 2a–d, shows a degree of Pd located on the external surface within PdNP/H-MZSM5, whilst the same is not apparent in Pd$_{imp}$/H-MZSM5. Consistent with our expected deposition locations within the two systems, Pd within Pd$_{imp}$/H-MZSM5 is more uniformly distributed through the meso and micropores, while Pd in PdNP/H-MZSM5 is compartmentalised at external sites and within directly accessible mesopores. Cross-section slices through the reconstructions (Fig. 2e, f and Supplementary Fig. 4), taken from the centre of the zeolite particle, reinforce our deduction that Pd in PdNP/H-MZSM5 is located within mesopores close to the external surface of the zeolite crystal, whereas Pd$_{imp}$/H-MZSM5 comprises Pd NPs within the interior region of the framework, with species present within the micropores and mesopores. For Pd residing within the micropores, the NPs can span multiple micropores or can induce the partial collapse of the localised framework. Thus, Pd NPs larger than a single micropore exists but are encapsulated within the micropore framework, i.e. accessible only via a micropore[44]. Identification of Pd in the same location in the cross-section slices above and below those in Fig. 2, as shown in Supplementary Fig. 4, verifies our identification of Pd within the tomogram and the corresponding slices.

## Mutually exclusive sites within hierarchical ZSM-5

Having developed a protocol to selectively deposit Pd NPs within mesopores of hierarchical ZSM-5, the attention can turn to the intrinsic Brønsted acid sites within ZSM-5. The aim to truly segregate the two different sites is complicated given the capacity for mesopores, in addition to micropores, to accommodate surface acid sites, both Brønsted and Lewis, the latter due to extra-framework aluminium species[48]. Therefore, removing such surface acidity within the mesopores is paramount to completely segregating two active species. The deployment of a bulky chelating agent, diethylenetriaminepentaacetic acid (DPTA), with a molecule diameter of ~0.9 nm (Supplementary Fig. 5) permits selective extraction of Al species (via complexation) from the mesopores/outer surface areas only[49,50], as depicted in Fig. 3a. Post-DPTA treatment (H-MZSM5-DA) crystallinity and porosity of the zeolite framework are unaffected, as evidenced by XRD and N$_2$ porosimetry (Supplementary Fig. 6 and Supplementary Table 1). Furthermore, this dealumination treatment restores Si:Al close to that of the parent ZSM-5, 34 (H-MZSM5-DA) and 40 (H-ZSM-5), an increase from 24 for H-MZSM5 (resulting from the initial desilication).

Acid-base titration using 2,6-ditertbutylpyridine (DTBPy), 3,5-dimethylbenzylamine (DBAM in acetone), and ethylenediamine (EDA) enables either partial or total titrations of the acid sites within the hierarchical zeolite catalysts due to steric limitations for DTBPy and DBAM relative to EDA. The large molecular dimension of DTBPy, kinetic diameter of 1.05 nm[51], limits its interaction to solely acid sites within the mesopores and on the external surface. Attenuated total

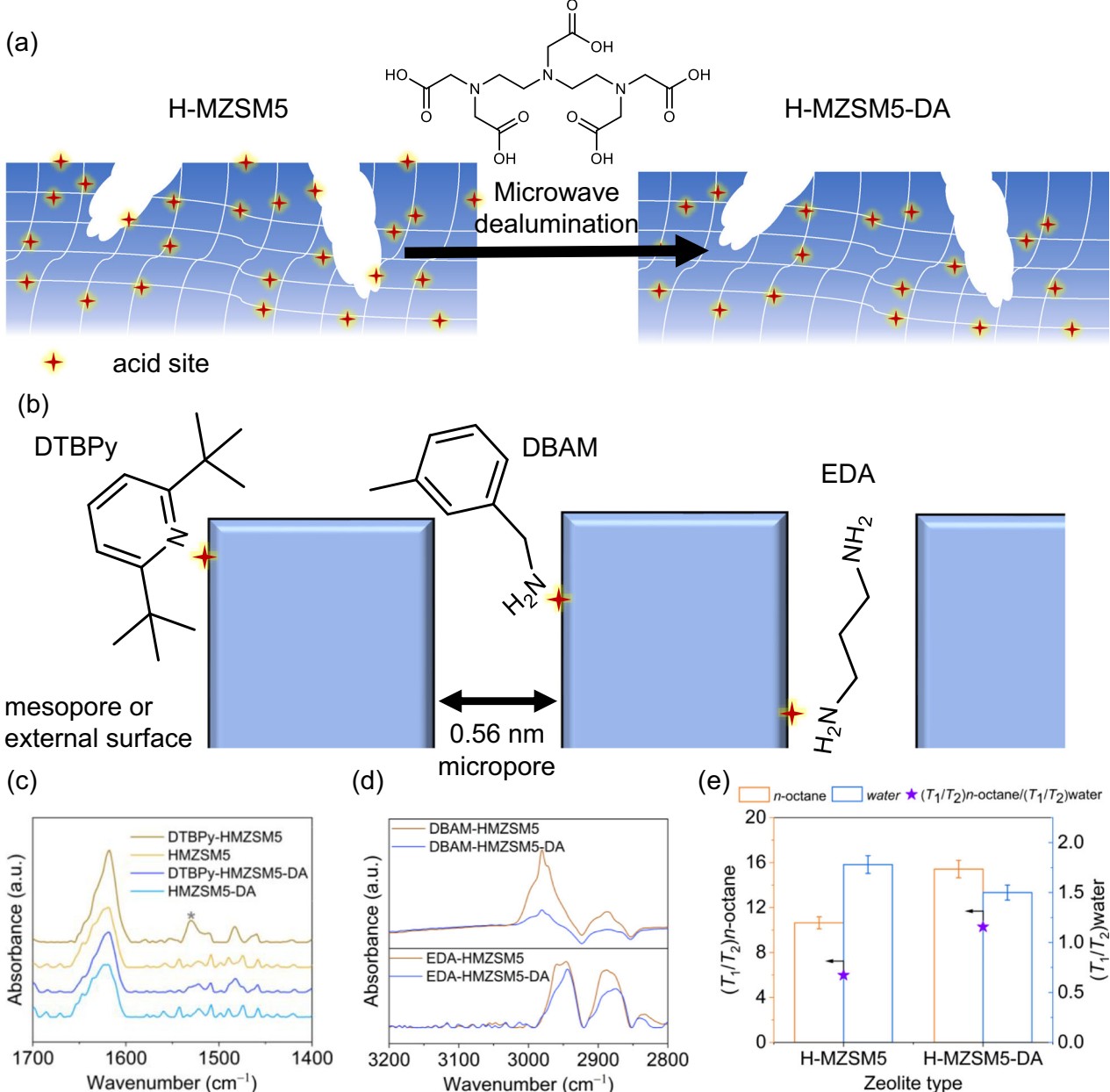

**Fig. 3 | Investigation of acid site location in H-MZSM5 and H-MZSM5-DA.**
**a** Schematic of H-MZSM5 dealumination. **b** Schematic of using 2,6-di-tert-butyl-pyridine (DTBPy), 3,5-dimethylbenzylamine (DBAM) and ethylenediamine (EDA) as probes to titrate acid sites. **c** ATR-IR spectra for the DTBPy-titrated H-MZSM5 and H-MZSM5-DA and their pure zeolite equivalents. **d** ATR-IR spectra for DBAM- and EDA-titrated H-MZSM5 and H-MZSM5-DA. **e** NMR relaxation $T_1/T_2$ ratios of $n$-octane and water in H-MZSM5 and H-MZSM5-DA and corresponding $(T_1/T_2)_{n\text{-octane}}/(T_1/T_2)_{water}$. The relative error on $T_1/T_2$ measurements is ~±5%.

reflection infrared (ATR-IR), shown in Fig. 3c, confirms DTBPy only adsorbs on H-MZSM5. The band at ~1530 cm$^{-1}$ is attributed to the DTBPyH$^+$ ion[52] from the interaction of DTBPy with Brønsted acid sites[53], thus validating the absence of acid sites at the external surface and within mesopores for the DPTA-treated zeolite (H-MZSM5-DA). In comparison, in addition to titrating external and mesopore-bound acid sites, DBAM, based on the pendant amine chain length, probes acid sites at a depth of up to ~0.3 nm into a micropore (Fig. 3b). CH stretching bands are observed in the region of 2800–3000 cm$^{-1}$ by ATR-IR, shown in Fig. 3d, with the intensity of these bands for H-MZSM5-DA greatly reduced relative to the parent material. This reduced acid site density is consistent with the successful removal of the mesopore and external surface acidity, with DBAM adsorbed at acid sites within the micropores close to the mesopores and external

surface, i.e. within the first zeolite cage from the mesopores or external surface. In contrast, EDA is unselectively adsorbed, titrating sites in the micropores, mesopores, and outer surfaces. CH stretch intensities reveal only a slight drop post-DPTA treatment (Fig. 3d), which again is consistent with the selective removal of sites from the mesopores and external surface only. Based on the combined results from the acid-base titrations, it can be concluded that DPTA treatment of H-MZSM-5 selectively extracts acidity from the mesopores and external surfaces only, with the characteristic micropore acidity of the zeolite, even in the micropores adjoining the mesopores and external zeolite surface, retained.

NH$_3$-TPD (Supplementary Fig. 6c) further confirms the decrease in total acidity. Weak and strong acid peak intensity, evaluated by desorption below and above 300 °C, respectively, reveals H-MZSM5-DA

possesses lower weak (0.219 vs 0.320 mmol g$^{-1}$) and strong acidity (0.340 vs 0.396 mmol g$^{-1}$), with the loss of weak site symptomatic of the removal of Al from outside the micropore framework. A side effect of removing acidity is the potential change in the hydrophobic-hydrophile nature of the material[54], with NMR relaxation used to probe internal hydrophobicity/hydrophilicity[55,56]. As shown in Fig. 3e, Supplementary Fig. 7, and Supplementary Table 3, T1 and T2 NMR relaxation times of water and n-octane in H-MZSM5 and H-MZSM5-DA probed variations in hydrophobicity. After post-treatment with DPTA, the zeolite framework becomes more hydrophobic, confirmed by a 16% reduction of the $T_1/T_2$ of water, i.e. 1.5 vs 1.8. Conversely, the interaction with the organic non-polar hydrocarbon probe, n-octane, is enhanced, with a $T_1/T_2$ ratio of 15.4 for H-MZSM5-DA vs 10.6 for H-MZSM5. The subsequent ratio of the $T_1/T_2$ values of n-octane to water is indicative of the relative surface affinity towards hydrocarbons relative to water[57,58]. As shown in Fig. 3e, DPTA removal of Al increases hydrophobicity (and correspondingly the affinity for hydrocarbons), which has been shown to be beneficial for catalytic reactions in which water is the by-product through its displacement from the catalyst[59–61]. The interaction of a model substrate and intermediate, 1-nonanoic acid and 1-nonanol, respectively, with the zeolites (H-MZSM5 and H-MZSM5-DA) can be considered to arise from a combination of the non-polar hydrocarbon chain and polar functional group, i.e. an intermediate of the two initial probes (water and n-octane) rather than one dominating over the other (shown in Supplementary Fig. 8 and Supplementary Table 3). Dealumination results in an increase in interaction for both model substrate and intermediate, reflecting a significant affinity between the hydrocarbon chain and the increased hydrophobic mesopore surface rather than a decreased interaction due to the repulsion of the polar functional group.

Having successfully extracted acidity from the mesopores and external surface to produce H-MZSM5-DA, the selective deposition of Pd, via the route deployed for PdNP/H-MSM5, yields a spatially orthogonal system with true active site compartmentalisation, i.e. acid sites only in micropores and Pd solely in mesopores and on external surfaces (PdNP/H-MSM5-DA). Supplementary Fig. 9 and Supplementary Table 1 reveal its analogous nature to its parent zeolite (where

applicable) and PdNP/H-MZSM-5, with equivalent porosity (surface areas, pore volumes and pore diameter), PdNP size (4.5 ± 0.9 nm from HAADF-STEM), and Pd:Si atomic ratios from surface, semi-bulk, and bulk analysis (0.0125 for XPS, 0.0118 for HAXPES, and 0.0048 for bulk analysis, to be compared with Fig. 1c) substantiating their comparable physicochemical nature. However, close inspection of the Pd *3d* XPS reveals a subtle shift to lower binding energy. Given the identical Pd particle sizes for PdNP/H-MSM5 and PdNP/H-MSM5-DA, this shift in energy must arise from the interaction with the acid site and is not a result of size effects. This shift results from interactions between the mesopore-located acid sites (both Brønsted and Lewis) and the Pd NPs within PdNP/H-MZSM-5. The acidity imparts an δ+ charge on the Pd NPs, and hence a higher binding energy[62,63]. Thus, removing these acid sites eradicates this influence on the electronic nature of the deposited Pd NPs, i.e. further evidence of the absence of mesopore acid sites in PdNP/H-MSM5-DA. XPS of Pd$_{imp}$/H-MZSM5 reveals a larger impact on Pd binding energy, attributed to the smaller PdNP size, here being both a particle size effect and a greater Pd-acid site interaction. PdNP$_{small}$/H-MZSM5 (Supplementary Fig. 9), prepared using pre-reduced Pd NPs with a diameter of only 2.7 ± 0.5 nm, shows an identical degree of mesopore acid site influence.

## Catalytic cascade HDO of Lauric acid to alkanes

We propose that the spatial segregation within the reported catalytic systems should be beneficial to a cascade hydrodeoxygenation that proceeds stepwise as follows: reduction (carboxylic acid to alcohol) in the mesopores over Pd, dehydration (alcohol to alkene) in the micropores over acid sites, and finally hydrogenation (alkene to alkane) over Pd in the mesopores, with the alkane product diffusing to the bulk reaction media, i.e. cascade intermediates are contained within the zeolite framework and subsequently consumed until the final product is generated. The stepwise cascade pathway is illustrated in Fig. 4, which is also observed for Ni/ZSM-5 and Mo/ZSM-22[64,65]. Observing cascade intermediate species is dependent on the catalyst configuration and species reactivity, e.g. carbonyl and alkene intermediates are often not observed due to rapid conversion[36,37], i.e. they are consumed before they can accumulate in the bulk solution.

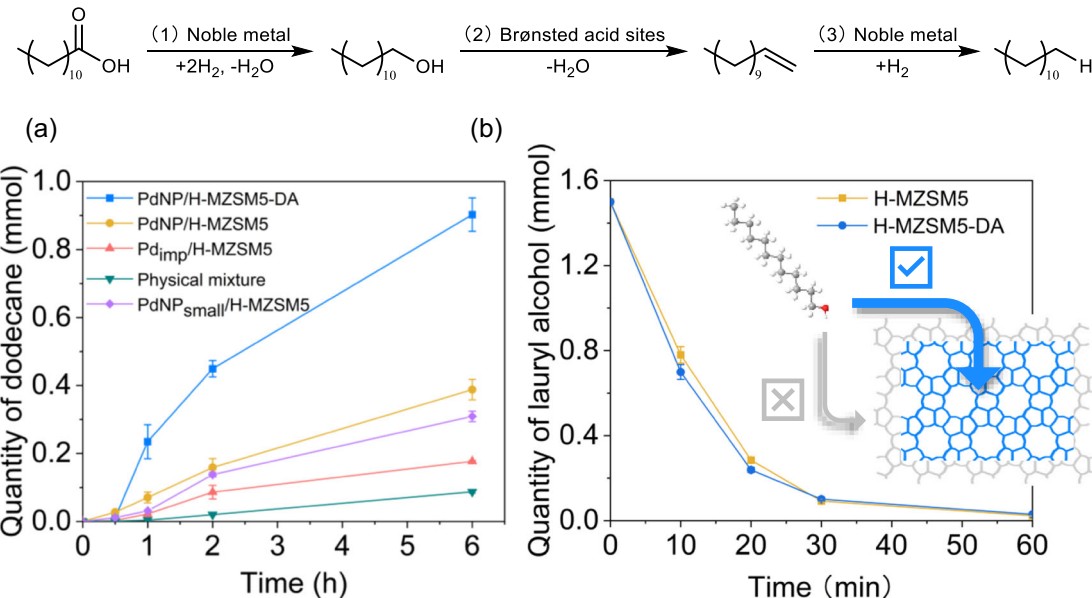

**Fig. 4 | HDO cascade reaction pathway of lauric acid to dodecane. a** HDO of lauric acid monitored from dodecane yield over PdNP/H-MZSM5-DA, PdNP/H-MZSM5, Pd$_{imp}$/H-MZSM5, PdNP$_{small}$/H-MZSM5 and physical mixture of H-MZSM5 + PdNP/Na-MZSM5. Reaction conditions: 100 mg 1 wt% Pd-doped catalyst, 300 mg lauric acid, 0.1 cm³ of nonane as the internal standard, 40 cm³ of hexane as the solvent, at 200 °C, 30 bar and 1000 rpm for 6 h. **b** Dehydration of lauryl alcohol of H-MZSM5 and H-MZSM5-DA to evaluate the rate of intermediate conversion, with the inset showing the location of dehydration reactions within the zeolite. Error bars in (**a**, **b**) correspond to the standard deviation of the average taken over at least two independent measurements.

**Table 1 | Lauric acid HDO performance over conventional prepared and spatial compartmentalised catalysts.[a]**

| Catalysts | Acid Conv., % | $C_{12}$ Sel., % | $C_{11}$ Sel., % | Productivity [b] ($mol_{C_{12}}$ $mol_{Pd}^{-1}$ $h^{-1}$) | TON[c] |
|---|---|---|---|---|---|
| H-MZSM5 | 0 | 0 | 0 | 0 | 0 |
| PdNP/Na-MZSM5 | 2.2 | 27.3 (alcohol) | 0 [d] | 0.05 (alcohol) | 29 |
| H-MZSM5 + PdNP/Na-MZSM5 [e] | 20.6 | 28.1 | 2.1 | 1.7 | 278 |
| PdNP/H-MZSM5 | 51.0 | 51.8 | 9.9 | 11.9 | 689 |
| Pd$_{imp}$/H-MZSM5 | 31.2 | 38.4 | 20.5 | 3.6 | 187 |
| PdNP/H-MZSM5-DA | 94.1 | 64.9 | 12.6 | 30.6 | 1272 |
| Pd$_{imp}$/H-MZSM5-DA | 70.9 | 39.9 | 18.2 | 11.3 | 426 |
| PdNP$_{small}$/H-MZSM5 | 50.9 | 40.4 | 14.9 | 9.9 | 688 |
| PdNP$_{small}$/H-MZSM5-DA[f] | 71.5 | 59.3 | 24.4 | 28.1 | 1205 |

[a]Reaction conditions: 100 mg 1 wt% Pd-doped catalyst, 300 mg lauric acid, 0.1 cm³ of nonane as the internal standard, 40 cm³ of hexane as the solvent, at 200 °C, 30 bar and 1000 rpm for 6 h.
[b]Production rate was calculated based on the yield of n-$C_{12}$ + iso-$C_{12}$ in 2 h, as shown in Fig. 4a. The unit is $mol_{dodecane}$ $mol_{Pd}^{-1}$ h⁻¹, in which the quantity of Pd is based on Pd loading from ICP.
[c]TON calculated based on moles of lauric acid converted at 6 h by surface Pd.
[d]The concentration of $C_{11}$ is below the GC detection limit.
[e]Physical mixture of 100 mg H-MZSM5 and 100 mg PdNP/Na-MZSM5.
[f]80 mg 1 wt% Pd-doped catalyst. The errors of conversion and selectivity are within ±3%, whereas yields and production rates are within ±5%.

HDO of lauric acid (dodecanoic acid) evaluates the effectiveness of PdNP/H-MZSM-5 and PdNP/H-MZSM5-DA relative to Pd$_{imp}$/H-MZSM5 and a physical mixture of monofunctional catalysts, H-MZSM5 (acid only) and PdNP/Na-MZSM5 (Pd only). The latter was prepared by applying the same synthesis strategy as PdNP/H-MZSM5 to the Na⁺ form equivalent zeolite, which possesses similar Pd properties (Supplementary Table 2 and Supplementary Fig. 10), albeit with a reduced Pd binding energy due to electron donation from the alkali metal[66]. Optimised reaction conditions are established on dodecane total yield (combination of n-dodecane, n-$C_{12}$, and iso-dodecane, iso-$C_{12}$), shown in Supplementary Table 4 and Supplementary Fig. 11. The process also yields undecane ($C_{11}$), through undesirable DCO$_x$, this being the main by-product detected (Supplementary Fig. 12), while trace amounts of lauryl laurate from the esterification reaction between the fatty alcohol and acid are also detected. The concentration of the latter does not increase significantly over the duration of the reaction, suggesting that its formation is a minor pathway. The impact of pressure shows at ≤20 bar, only selectivity is influenced whilst conversion is comparable, whereas increasing pressure to 30 bar drives up conversion while maintaining higher $C_{12}$ selectivity. The impact of stirring rate shows no effect, as presented in Supplementary Fig. 11, ruling out bulk mass diffusion effects.

The synergy between the two sites, viz. metal and acid, is paramount to enabling HDO. Control experiments over one-site-only systems, i.e. H-MZSM-5 and PdNP/Na-MZSM5 (Table 1), reveal acidity alone is unable to convert the starting substrate. In comparison, the sole deployment of Pd via PdNP/Na-MZSM5 produces only the first intermediate product (1-dodecanol) at low levels, consistent with the initial reduction step of the cascade[67]. A physical mixture of these two materials does produce a $C_{12}$ alkane yield (Fig. 4a), thus confirming the criticality of a union of metal and acid sites for the catalytic cascade HDO. While also demonstrating the power of coupling these two sites in driving the cascade reaction forward, i.e. consumption of the alcohol by the acid sites drives the equilibrium increasing acid reduction, hence higher lauric acid conversion relative to PdNP/Na-MZSM5. However, a physical mixture does not permit governance over the order in which the different catalytic active species are encountered and is thus far from optimal[9,24–26]. When this reaction parameter is controlled by deploying PdNP/H-MZSM5, it yields a significant enhancement, with dodecane yield increasing by ~20% and a sevenfold increase in production rate, a direct consequence of dictating the reaction sequence of the cascade. Decreasing PdNP size whilst retaining spatial compartmentalisation, i.e. PdNP$_{small}$/H-MZSM5, does not reveal an anticipated kinetic enhancement from the increased Pd dispersion, i.e. faster lauric acid conversion from increased surface Pd

sites. In fact, it results in reduced $C_{12}$ selectivity. Thus, decreasing Pd particle size harms overall HDO performance. This correlation between HDO performance and PdNP size is further evident from the further reduced HDO performance of Pd$_{imp}$/H-MZSM5, with a greater impact, at least partially, due to accessibility, i.e. Pd in micropore being less accessible. The drop in the capacity towards HDO, including lauric acid conversion, is in conjunction with an increase in $C_{11}$ selectivity. The latter arising from the proximity, i.e. adjacent, of Pd and acid sites, with their interaction favouring decarboxylation over HDO[68], and the decrease in Pd size[69]. Thus, decreasing Pd particles is detrimental to overall HDO performance, exacerbated by contiguous Pd and acid sites, i.e. when co-located within the micropores or mesopores of the zeolite framework. However, comparing HDO performance for materials with comparable average particle sizes, impregnation vs small Pd NPs, further highlights the benefit of spatial segregation on the HDO cascade process, with reduced $C_{11}$ formation resulting from suppressing interaction between Pd and acid sites.

While it is expected that compartmentalisation of Pd into the mesopores of PdNP/H-MZSM5 is likely beneficial for substrate diffusion, given the dimension of lauric acid (Supplementary Fig. 13), we acknowledge that a degree (albeit minor based on the NH₃-TPD) of acid sites co-exist within the mesopores. This raises the question regarding which sites are responsible for the alcohol dehydration step, i.e. acid sites within mesopores or micropores. The dehydration of lauryl alcohol over H-MZSM5 and H-MZSM5-DA (acid only zeolites), as shown in Fig. 4b, reveals almost identical reaction rates (1.62 and 1.74 mmol h⁻¹, respectively), confirming that Brønsted acid sites within the micropores of the zeolite framework drive dehydration. Thus, Pd and acid sites separated at the nanoscale operate individually for their respective steps within the cascade process but collaborate for overall optimal HDO performance. Furthermore, as shown in Fig. 4b and Supplementary Fig. 14, the rate of dehydration is significantly greater than that of lauric acid consumption (~0.3 mmol h⁻¹) for PdNP/H-MZSM5. The initial reduction step of the acid to the alcohol is, therefore, slower than the subsequent alcohol dehydration step, which is further supported given that cascade intermediate species are not observed, i.e. intermediates are consumed rapidly within the pore architecture of the bifunctional catalysts as substrates for the subsequent step in the cascade before having time to diffuse into the bulk reaction media. Likewise, alkene hydrogenation is another facile step in the cascade and is thus consumed before building up in the reaction media[36,37].

HDO activity further escalates with true active site compartmentalisation, with PdNP/H-MZSM5-DA showing the highest rate of dodecane production (30.6 $mol_{dodecane}$ $mol_{Pd}^{-1}$ h⁻¹), a 2.5-fold increase on

PdNP/H-MZSM5, and $C_{12}$ selectivity. Furthermore, the rate of lauric acid consumption increases by a comparable factor (~2). As previously observed in the non-dealuminated system, the incorporation of small Pd NPs (PdNP$_{small}$/H-MZSM5-DA) with a greater number of surface sites is not conducive to improved activity relative to its larger NP equivalent (PdNP/H-MZSM5-DA). Likewise, it induces increasing $C_{11}$ formation, although now solely due to reduced NP size[69]. That said, achieving true compartmentalisation once again is highly beneficial, as evident by the amplified activity and conversion relative to the impregnated and non-dealuminated equivalents (Pd$_{imp}$/H-MZSM5-DA and PdNP$_{small}$/H-MZSM5, respectively). While PdNP$_{small}$/H-MZSM5 does give rise to a significant increase in performance relative to Pd$_{imp}$/H-MZSM5, it does not reach the level of PdNP/H-MZSM5-DA or PdNP$_{small}$/H-MZSM5-DA, confirming the benefit of spatial orthogonality of active species on catalytic HDO. The origin of the enhancement from spatial separation of the active species resulting from the removal of the mesopore acidity is attributed to the coeffect of; (i) the elimination of the undesirable interaction between acid sites and Pd NPs, and (ii) the increased mesopore hydrophobicity, which aids the expulsion of water (the by-product of acid reduction), driving the localised equilibrium forward increasing alcohol production. Furthermore, $H_2$ dissociation on the PGM sites has been shown to be hindered by the presence of water[70]. Concurrently, increasing hydrophobicity within PdNP/H-MZSM5-DA does not negatively impact interaction with the fatty acid (discussed above) and, thus, would not negatively impact cascade process efficiency. The inherent benefit of active site spatial compartmentalisation within PdNP/H-MZSM5-DA is further evident from its capacity for HDO of other fatty acids, including palmitic (C16) and stearic (C18) (Supplementary Table 5).

Catalyst recycling studies show that PdNP/H-MZSM5-DA exhibits good recyclability (Supplementary Figs. 15, 16) without requiring reactivation between runs, whilst PdNP/H-MZSM5 shows reasonable recyclability (TON drop from 688 to 574 over three cycles). In contrast, PdNP/H-ZSM-5, comprising Pd NPs deposited on the external surface of micropore-only ZSM-5, exhibits poor catalyst recyclability, deactivating by ~90% over only two runs. HAADF-STEM imaging confirms Pd agglomeration is the root cause of this, which arises from the lack of mesopore confinement of the Pd NPs. Thus, the catalyst configuration possessed by PdNP/H-MZSM5 and PdNP/H-MZSM5-DA bestows yet another highly beneficial asset on our catalytic materials. Moreover, the formation of coke, another common deactivation pathway in zeolite catalysis, can be ruled out. Thermogravimetric analysis of PdNP/H-MZSM5-DA (Supplementary Fig. 17) shows mass losses only below 300 °C, which we attributed to arise from the desorption of adsorbed substrate (dodecanoic acid b.p. 299 °C), intermediates (dodecanal b.p. 242 °C, dodecanol b.p. 259 °C, and dodecene b.p. 213 °C), and product (dodecane b.p. 216 °C), whereas combustion of coke species requires temperatures ≥500 °C. So, while species are adsorbed, which accounts for some of the mass balance of the process (the remaining missing mass may reflect low solubility and precipitation of the substrate during sampling), it is reasonable to assume they remain as species that partake in the cascade process. This varies from studies into unsaturated triglycerides where coking and oligomerisation are apparent due to reactivity from unsaturation in the initial substrate[71].

To further explore this approach and evaluate the impact of intrinsic zeolite acidity, the optimal synthetic route was applied to ZSM-5 with Si:Al ratios of 25 and 15. Increasing the inherent acidity of the catalytic system shows little effect on overall conversion (Supplementary Table 6), note the higher substrate concentration and lower metal loading, with only a decrease observed at the lowest Si:Al ratio. In contrast, selectivity towards HDO is compromised by the increasing acidity, which negatively impacts $C_{12}$ alkane productivity. Furthermore, switching to other zeolite frameworks, namely USY and Beta, further demonstrates the generic nature of the approach for catalyst

synthesis. While resulting in superior acid conversion, it is at the expense of $C_{12}$ alkane selectivity.

In conclusion, the successfully developed a hierarchical ZSM-5 catalyst involving spatial compartmentalisation of Pd (mesopores) and acid (micropores) sites, and its influence over the one-pot HDO of lauric acid to dodecane has been demonstrated. Compared with conventional catalyst production, via impregnating with Pd salts, Pd within the spatially separated catalysts resides within mesopores and on the outer surfaces, as evidenced by electron tomography. This facilitates greater accessibility for the initial acid reduction step of the cascade, whereas the dehydration of lauric acid, the second step in the cascade, is shown to occur over acid sites within the micropores of hierarchical ZSM-5, potentially within micropores bordering mesopores. Furthermore, the segregation of Pd and acid sites impedes undesirable side reactions, i.e. decarboxylation and decarbonylation, while the elimination of mesopore acidity, tunes the hydrophobicity of the mesopores and interactions between Pd and acid sites, which further accelerates overall HDO performance. Moreover, this approach prohibits Pd blocking/binding to the acid sites, another undesirable aspect of producing such bifunctional materials. However, we do point out that here, within our Pd$_{imp}$/H-MZSM5, the acid site of the parent zeolite significantly outweighs the amount of Pd impregnated, and thus, the resulting surface Pd atoms could, in theory, only block a maximum of 3.5% of the acid sites present, i.e. the impact of such an effect can be considered inconsequential.

The benefit of employing a spatially orthogonal catalyst, through active site compartmentalisation, for lauric acid HDO is further evident when it is benchmarked against the current state-of-the-art HDO catalysts (Supplementary Table 7), in which Pd is deployed either as an active species, promotor or stabiliser. Compared with mono-metallic Pd systems[29], the higher selectivity of the catalyst reported here drives greater HDO performance, as while higher reaction temperatures deployed in the literature enhance conversion, this is at the expense of favouring DCO$_x$ over HDO. Furthermore, the benefit of spatial orthogonality within PdNP/H-MZSM5-DA is further evident from its superior performance to Pd supported randomly on acidic alumina silicates (Pd/Al-SBA-15, Pd@Al-SiO$_2$)[29,33], and zeotype materials (Pd/SAPO-31)[29], which again is due to superior selectivity. However, relative to acidity co-deposited on silicas (Pd-Nb$_2$O$_5$/SiO$_2$ and Pd/HPA-SiO$_2$)[34,72] and bimetallic systems (Pd/CuZnAl + ZSM-5 and Pd-Re/C + zeolite A)[40,43] it is the greater mol normalised activity of PdNP/H-MZSM5-DA that gives rise to greater HDO productivity. Comparisons are also made with Pt-based systems, with either comparable (Pt/Nb$_2$O$_5$) or superior (Pt/ZSM-5, Pt/TiO$_2$, Pt-Re/TiO$_2$) alkane productivities observed. Therefore, it is a combination of activity and selectivity that results in the optimal HDO performance of PdNP/H-MZSM5-DA, which equals or outperforms the current literature demonstrating the advantage of nanoscale active species segregation in catalytic cascade HDO processes. Thus, this study should inspire future HDO catalyst development while also highlighting the potential of spatial separation of active species within multifunctional materials for one-pot cascade reactions, i.e. catalysts that can facilitate the combination of multiple steps of a chemical production process within a single reactor. Furthermore, it demonstrates a generic synthesis approach for the production of bifunctional materials, in which different functionalities are compartmentalised within two distinct regions.

## Methods
### Chemicals and materials
ZSM-5 zeolite Si:Al 40 (Zeolyst International as ammonia form, CBV 8014), ZSM-5 zeolites Si:Al 15 and 25 (Fisher Scientific), sodium hydroxide (NaOH, ACROS Organics, ≥99%), ammonium nitrate (NH$_4$NO$_3$, ACROS Organics, ≥98%), diethylenetriaminepentaacetic acid (DPTA, Sigma Aldrich, ≥99%), tetraaminepalladium(II) nitrate solution (Pd(NH$_3$)$_4$(NO$_3$)$_2$, Sigma Aldrich, 10 wt.%), ethylene glycol (EG, Fisher

Scientific, ≥99%), poly(vinyl pyrrolidone) (PVP, Sigma Aldrich, $M_w$ = ~55,000), sodium tetrachloropalladate(II) ($Na_2PdCl_4$, Sigma Aldrich, ≥98%), 2,6-di-tert-butylpyridine (DTBPy, Sigma Aldrich, ≥97%), 3,5-dimethylbenzylamine (DBAM, Sigma Aldrich, ≥98%), ethylenediamine (EDA, Fluorochem, 99.0%), hexane ($CH_3(CH_2)_4CH_3$, Sigma Aldrich, ≥95%), nonane ($CH_3(CH_2)_7CH_3$, Sigma Aldrich, ≥99%), lauric acid ($CH_3(CH_2)_{10}COOH$, Sigma Aldrich, ≥98%), and lauryl alcohol ($CH_3(CH_2)_{11}OH$, Sigma Aldrich, ≥98%). All chemicals were used as received.

### Preparation of catalysts

Hierarchical ZSM-5 was prepared by alkaline desilication as reported by ref. [73]. ZSM-5 Si:Al 40 was calcined at 550 °C for 5 h (ramp rate 5 °C min$^{-1}$) to convert to the H-form (denoted H-ZSM-5). H-ZSM-5 (6 g) was vigorously stirred in aq. NaOH (0.2 M, 180 cm$^3$) at 65 °C for 0.5 h and then washed with deionised water to neutral pH. Ion exchange was carried out using aq. $NH_4NO_3$ (0.1 M, 180 cm$^3$) at 80 °C for 3 h in triplicate. The solid was isolated by centrifugation (relative centrifugal force 3000 G, 10 min), dried at 100 °C for 6 h, and calcined in air at 550 °C for 5 h (ramp rate 5 °C min$^{-1}$) to generate the H-form hierarchical ZSM-5 (denoted H-MZSM5). Na$^+$-form hierarchical ZSM-5 (sample denoted Na-MZSM5) was prepared as above, with the ion exchange step omitted.

Mesopore and external surface acid sites were removed from H-MZSM5 by microwave-assisted chelation[50]. H-MZSM5 (1 g) was dispersed in aq. DPTA (0.16 M, 15 cm$^3$) by stirring at room temperature (20 °C). The solution was heated to 100 °C (ramp rate 25 °C min$^{-1}$) and treated isothermally for 0.25 h within an Anton Paar Monowave 400 microwave reactor, followed by washing, drying and calcination at 550 °C for 5 h (ramp rate 5 °C min$^{-1}$) to give the sample H-MZSM5-DA.

Near monodisperse Pd NPs of ~4.5 nm were prepared using a procedure reported by ref. [74]. PVP (450 mg) was dissolved in EG (20 cm$^3$) at 160 °C. A second solution containing $Na_2PdCl_4$ (155 mg) dissolved in EG (10 cm$^3$) at 60 °C was added to the PVP-EG solution at 160 °C under stirring. After 3 h, acetone (60 cm$^3$) was charged into the solution to precipitate Pd NPs. The solid was separated by centrifugation and washed with water and acetone (1:3, v/v) mixture in triplicate. Pd NPs were dispersed in water (30 cm$^3$) to yield a colloidal solution. Pd NPs of ~2.5 nm were prepared using an increased mass of PVP (1550 mg).

Deposition of Pd NPs on H-MZSM5 was conducted by NP impregnation. ~4.5 nm PdNP colloid (6 cm$^3$) was mixed with H-MZM5 (1 g) to give a nominal 1 wt% Pd loading. The slurry was stirred at 20 °C for 18 h, then heated to 50 °C to isolate a solid. The solid was calcined at 400 °C for 2 h (ramp rate 1 °C min$^{-1}$) and reduced at 200 °C under $H_2$ for 2 h (ramp rate 5 °C min$^{-1}$, $H_2$ 100%, flow rate 100 cm$^3$ min$^{-1}$) to yield PdNP/H-MZSM5. An identical approach was deployed for the following supports: H-ZSM-5, Na-ZSM-5, and H-MZSM5-DA, to produce catalysts denoted as PdNP/H-ZSM-5, PdNP/Na-ZSM-5, and PdNP/H-MZSM5-DA, respectively. PdNP$_{small}$/H-MZSM5 and PdNP$_{small}$/H-MZSM5-DA were prepared as above but using the ~2.5 nm PdNP colloid.

Control catalysts were prepared via conventional wet impregnation of a Pd salt. H-MZSM5 (or H-MZSM5-DA) (1 g) was suspended in an aq. $Pd(NH_3)_4(NO_3)_2$ (3 cm$^3$) solution with Pd concentration adjusted for a nominal 1 wt% Pd loading. The slurry was stirred at 20 °C for 18 h, then heated to 50 °C to isolate a solid. The solid was calcined at 500 °C for 2 h (ramp rate 1 °C min$^{-1}$) and reduced under $H_2$ at 400 °C for 2 h (ramp rate 5 °C min$^{-1}$, $H_2$ 100%, flow rate 100 cm$^3$ min$^{-1}$). The obtained catalyst is denoted $Pd_{imp}$/H-MZSM5 ($Pd_{imp}$/H-MZSM5-DA).

### Characterisation of materials

Powder X-ray diffraction (XRD) was conducted on a Philips X'Pert-PRO theta-theta PW3050/60 diffractometer with a PW3064 sample spinner and X'Celerator 1-D detector (2.122° active length) in Bragg-Brentano geometry using a copper line focused X-ray tube with Ni Kβ absorber (0.02 mm; Kβ = 1.392250 Å) Kα radiation (Kα$_1$ = 1.540598 Å, Kα$_2$ = 1.544426 Å, Kα ratio 0.5, Kα$_{ave}$ = 1.541874 Å). Diffraction patterns were collected from 5 to 75 ° 2 theta at 0.0334 ° step and 1.7 s step$^{-1}$. The relative crystallinity of the zeolite materials was determined by a standard integrated peak area method using 22.5–25.0° 2 theta (Eq. S1)[75].

Nitrogen adsorption/desorption isotherms were measured on a Micromeritics ASAP 2020 porosimeter. The materials were degassed under vacuum at 350 °C for 12 h before nitrogen adsorption at −196 °C. Brunauer–Emmett–Teller (BET) surface areas ($S_{BET}$) were calculated over the relative pressure range of 0.03–0.30, the specific micropore surface area ($S_{micro}$) and volume ($V_{micro}$) were calculated over the relative pressure range of 0.3–0.5 using the $t$-plot method. Pore size distribution was calculated by the Barrett–Joyner-Halenda (BJH) method with Faas correction using the adsorption branch of the isotherms.

Ammonia temperature-programmed desorption (NH$_3$-TPD) was conducted on a Quantachrome ChemBET 3000 system. Catalysts (~50 mg) were pre-treated at 550 °C (ramp rate 5 °C min$^{-1}$) for 1 h under He. The sample was cooled to 100 °C and exposed to a NH$_3$/He (1:9 vol/vol, 30 cm$^3$ min$^{-1}$) gas mix for 1.5 h, followed by He purge (60 cm$^3$ min$^{-1}$) for 2 h to remove physisorbed NH$_3$. NH$_3$-TPD was performed from 100 to 600 °C (ramp rate 10 °C min$^{-1}$) under He (30 cm$^3$ min$^{-1}$), with desorbed NH$_3$ monitored by thermal conductivity. CO pulse chemisorption was performed using a Quantachrome ChemBET 3000 system. Samples were degassed at 150 °C for 1 h under He (20 cm$^3$ min$^{-1}$) before reduction at 100 °C for 1 h under pure H$_2$ (20 cm$^3$ min$^{-1}$). A milder reduction condition was used to avoid additional particle sintering. CO chemisorption was conducted at RT, with dispersion measurements evaluated based on CO Pd stoichiometry of 1 to 2[47].

X-ray photoelectron spectroscopy (XPS) was performed using a Kratos Axis Ultra Hybrid spectrometer with monochromated Al K$_\alpha$ radiation (1486.6 eV, 10 mA emission at 150 W, spot size 300 × 700 μm) at a base vacuum pressure of ~5 × 10$^{-9}$ mbar. Charge neutralisation was achieved using a low-energy electron flood gun. Pass energies of 80 and 20 eV were used for the survey and high-resolution scans. The total energy resolution of high-resolution scans was 0.5 eV, which is limited by the width of the emission line from the sample, the analyser, and a Gaussian and Lorentzian contribution to the lineshape. Hard X-ray photoelectron spectroscopy (HAXPES) was performed on a Scienta Omicron GmbH HAXPES-Lab using monochromated Ga K$_\alpha$ X-ray radiation (9252 eV, 3.57 mA emission at 250 W, micro-focussed to 50 μm) and an EW-4000 high voltage electron energy analyser[76,77]. Measurements were made at a base vacuum pressure of ~5 × 10$^{-10}$ mbar, with an entrance slit width of 0.8 mm. Pass energies of 500 and 200 eV were used for the survey and core level spectra, respectively, with total energy resolutions of 1.2 and 0.6 eV. Charge neutralisation was achieved using a PreVac FS40A low-energy electron flood source. In all tests, binding energy scale calibration was performed using Si $2p$ photoelectron peak at 103.4 eV. Analysis and curve fitting was performed using CasaXPS, with a Shirley background and an asymmetric lineshape of LA(1.9,7,2) for Pd(0) and a symmetrical line of LA(1.53,243) for all other components[78].

Transmission electron microscopy (TEM) was conducted with an FEI Tecnai F30 FEG-AEM TEM operating at 300 kV. High-angle annular dark-field scanning transmission electron microscopy (HAADF-STEM) images were recorded on an FEI Titan3 Themis G2 operated at an accelerating voltage of 300 kV, equipped with a Schottky field-emission gun (X-FEG) operating at an extraction voltage of 4.5 kV, a monochromator (energy spread ~0.25 eV) and an FEI Super-X 4-detector EDX system. The HAADF-STEM tilt series were performed using the FEI STEM Tomography set-up with images collected at 5° intervals over a ±70° range. The tilt series images were aligned and reconstructed in Inspect 3D (version 9.5.0) using the simultaneous

iteration reconstruction technique (SIRT). The reconstructed model was then visualised using Imaris to obtain the cross-section images and generate three-dimensional surfaces to aid in visualising the position of the nanoparticles.

Attenuated total internal reflection infrared spectroscopy (ATR-IR) was conducted on a Bruker Vertex 70 spectrometer with a deuterated triglycine sulfate detector and platinum ATR accessory. Samples were measured at room temperature under air. Spectra of adsorbed amine probe molecules (DTBPy, DBAM, and EDA) are an average of 64 scans with a resolution of 4 cm$^{-1}$, with air recorded as a background.

Nuclear magnetic resonance (NMR) relaxation measurements were carried out in a Magritek SpinSolve benchtop NMR spectrometer operating at a $^1H$ frequency of 43 MHz. $T_1$ relaxation experiments were performed using an inversion recovery pulse sequence[79], acquiring 16 experimental points for each experiment with time delay values between 1 ms and 1000 ms, 16–64 scans and a repetition time of 1000 ms. $T_2$ relaxation experiments were performed with the CPMG (Carr Purcell Meiboom Gill) pulse sequence[80], with an echo time of 250 µs, using 16 steps with a number of echoes per step varying in the range 30–70, 64–128 scans and a repetition time of 1000 ms. Samples were prepared by soaking the solid in the relevant probe liquid for 48 h. Excess external liquid was removed by placing the solid on pre-soaked filter paper. The solids were placed in 5 mm NMR glass tubes, and measurements were made at 20 °C ± 0.5 °C under atmospheric pressure.

Thermogravimetric analysis (TGA) was conducted on a TA Instruments TGA 550. TGA analysis was conducted under flowing air (40 cm$^3$ ml$^{-1}$) from room temperature to 800 °C (ramp rate 10 °C min$^{-1}$).

## Catalytic testing

Catalyst (-0.1 g) was charged into hexane (20 cm$^3$) in a 100 cm$^3$ Parr 4598 autoclave reactor and reduced under H$_2$ (30 bar) at 200 °C for 2 h with stirring (at 1000 RPM). After reduction, the reactor was purged to 5 bar and depressurised five times before the addition of lauric acid (0.3 g) in hexane (20 cm$^3$) and nonane (0.1 cm$^3$) as an internal standard. The system was pressurised with H$_2$ and heated to the desired reaction temperature. Once at temperature, reactions were initiated by agitation, with aliquots (<1 cm$^3$) extracted for reaction profiling, with analysis by gas chromatography. It is important to note that due to the potential for precipitation of the fatty acid substrate at room temperature, sampling should be conducted at the reaction temperature, and the sampling system should be flushed prior to sample collection to avoid carryover. Loss of the substrate during sampling will impact solely on the conversion reported. There are no issues with regard to product solubility. The dehydration of lauryl alcohol was conducted under identical conditions, with the omission of the pre-treatment. For catalyst recycle studies, additional lauric acid (0.3 g), hexane (3 cm$^3$) and nonane (0.015 cm$^3$) were added into the reactor without discharging the previous reaction media[81]. The system was returned to reaction conditions to commence the subsequent catalytic cycle.

## Data availability

Raw data that further supports and underpins the manuscript can be found in the supplementary information, with all raw data available via the online digital repository Figshare https://doi.org/10.48420/22722229 or from the corresponding authors upon reasonable request.

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

## Acknowledgements

This project has received funding from the European Union's Horizon 2020 research and innovation programme under grant agreement No 872102. M.H. thanks the China Scholarship Council (CSC, file no. 201806340224) for supporting her research. B.Q. thanks to the China Scholarship Council (CSC, file no. 202006240076) and the University of Manchester for the joint PhD studentship to support her research. X.O. thanks the Zhejiang Provincial Natural Science Foundation for funding (LQ23B060005). U.K. thanks the Punjab Educational Endowment Fund (PEEF). B.S. thanks the EPSRC grants EP/R00661X/1, EP/P025021/1, and EP/P025498/1. X.F. and Y.J. acknowledge the Key Project on Intergovernmental International Science, Technology and Innovation (STI) Cooperation/STI Cooperation with Hong Kong, Macao and Taiwan of China's National Key R&D Programme (2019YFE0123200) and the National Natural Science Foundation of China (No. 22378407) for supporting the collaboration. C.M.A.P. thanks the UK Catalysis Hub for the resources and support provided via our membership of the UK Catalysis Hub Consortium and funded by EPSRC grants EP/R026939/1 and EP/R027129/1. The authors thank the EPSRC grant EP/S021531/1 for TEM analysis at The University of Manchester. C.D. and X.F. also acknowledge the EPSRC (grant no. EP/V026089/1) for supporting the research.

## Author contributions

C.M.A.P. and X.F. conceived the work. C.M.A.P., X.F. and S.D. planned the experiments. S.D. synthesised materials under guidance from X.F. and C.M.A.P. S.D., Y.P. and U.K. performed catalytic testing. S.D., D.L.F.A., M.H., B.Q., X.O., B.S., C.D., X.Z., Y.J. and N.H. undertook materials characterisation. S.D., C.D., N.H., B.S., X.F., C.T., U.K. and C.M.A.P. analysed the data. S.D., X.F. and C.M.A.P. wrote the manuscript, with contributions from all other co-authors.

## Competing interests

The authors declare no competing interests.
