## [Peer Review File · Nature Communications]

Spatial segregation of catalytic sites within Pd doped H-ZSM-5 for fatty acids hydrodeoxygenation to alkanesEditorial Note: This manuscript has been previously reviewed at another journal. This document only contains reviewer comments and rebuttal letters for versions considered at Nature Communications.

REVIEWER COMMENTS

Reviewer #2 (Remarks to the Author):

Ding et al. reported a bifunctional Pd/H-meso-ZSM-5 with Pd NPs selectively placed in mesopores, which exhibited superior performance in hydrodeoxygenation of lauric acid in comparison to the one with small Pd NPs in zeolite crystals. This finding is different from the general understanding of the closer the better. The size of Pd NPs and their location along mesoporous zeolite have well been characterized. Generally, this manuscript is well-written and organized. However, I cannot suggest its publication at this stage because of the following concerns.

- (1) The findings of this work about the spatial organization of functional sites may benefit the development and production of multifunctional nanomaterials, but the performance improvement in HDO of lauric acid is not that significant compared with the literature (Catal. Sci. Technol., 2014, 4, 3705–3712), and even much lower. A comparison with published results should be made. Besides, based on the literature I looked through, it seems that the Pt-based catalysts are better than Pd-based catalysts and the price of Pt is even lower than that of Pd. The author should clarify why Pd was chosen for this material.
- (2) The dehydration of alcohol to alkene is kinetically feasible, not a rate-determining step. Why H-ZSM-5 with strong acid sites was employed. The investigation of the zeolite topology effect and support effect should be made to show the advantage of H-ZSM-5.
- (3) The Pd size of the optimum PdNP/H-MZSM5 catalyst is about 4.4 nm with a Pd dispersion of 26.5%, which means the utilization of Pd is not very high. The particle size effect or structure sensitivity of Pd NPs should be investigated. At least, PdNPsmall/H-MZSM5-DA should be included.
- (4) The authors stated that mesopore hydrophobicity aids the expulsion of water (the by-product of acid reduction), thus increasing alcohol production. However, if this is true, the hydrophobicity will also suppress the diffusion of lauric acid to Pd NPs, which are deposited on hydrophobic surfaces.
- (5) The title is too big based on the results in the manuscript. The authors only used lauric acid as a probe molecular for fatty acids. Some other fatty acids with different carbon numbers should be performed.

Reviewer #3 (Remarks to the Author):

Manuscript review comments

Review on the article: "Spatial segregation of catalytic sites within Pd doped H-ZSM-5 for fatty acids hydrodeoxygenation to alkanes". The authors have achieved commendable activity through accessibility of Pd active sites. Beyond that the article is well written and offers some excellent microscopy results and a very interesting approach to determination of the acid sites accessibility and strength. However, I think there is a fundamental issue with this catalyst design idea. The separation of acid and metal sites in such setup is detrimental to catalyst stability. The way this setup differs from the hydroisomerization where the approach works¹ is that the acidic sites here can form their own alkenes via the dehydration of the alcohols which enables very high potential for all the reactions that are

catalysed from the protonation of the double bonds – oligomerization, coking and so on. Additionally, these reactions have a concentration (second order or higher) and temperature dependence therefore the issue would increase exponentially with the reaction feed concentration increase and with temperature increase – which are both crucial for the consideration of the industrial use. The issue of acidity and high degree of unsaturation was addressed a few times in the literature including 2,3. The stability and the combination of acidic and hydrogenation function in proximity are the reason why transition metal sulphides (NiMoS / CoMoS) are used in industrial applications 4. Furthermore, I believe the superior activity is the result of the combination increase in the pore size⁵, Pd nanoparticles size and increased HDO selectivity which in turn prevents “CO poisoning”⁶ and reactions such as methanation due to better dehydration activity of the catalyst – the acid sites not being blocked by the Pd deposition in the Pd/MZSM5-DA case. Based on these reasons, my conclusions on the presented data would be that the sites still work together while not blocking each other which happens with the impregnation methods and that the key to their activity is in accessibility rather than separation and the data presented has not convinced me otherwise. Notably, there is no data on the acidity of the metal deposited catalysts which may vary significantly and guide the activity of the catalyst. Regardless of that, I believe that article should be published after the issue is described throughout the manuscript and other corrections are applied due to high activity, an interesting approach and overall good presentation of the results.

Comments

“Hierarchical mesoporous-microporous ZSM-5 zeolite is deployed as both catalysts, given its inherent acidity, and catalyst support to host deposited metal sites.” – As this is indeed true, it is unusual that related previous studies are not that well reviewed, as there are quite some existing.

Aspects to thus be considered:

- the hydrocracking, hydrogenation and hydro-deoxygenation of bio-based fatty acids, esters and glycerides, considering mechanisms, kinetics and transport phenomena resistances
- the conversion of the bio-based fatty acids over bi-functional Ni/ZSM-5 catalysts, considering the effect of stoichiometric Ni/Al molar ratio
- the mechanism, ab initio calculations and micro-kinetics of the straight chain alcohol, ether, aldehyde, bio-based carboxylic acid and ester hydro-deoxygenation reactions over Mo–Ni catalysts

The reaction scheme (Scheme 1) lacks a hydrogen molecule, two H₂ molecules are needed to go from fatty acid to alcohol, usually the hydrogenation of fatty aldehydes is very fast over noble metals thus those can not be seen on GC-MS.

The supplementary Fig. 11, applying a longer analysis method might unravel some esterification product in the form of lauryl laurate. It's a common product of weaker acid sites, especially at lower temperatures in the reactions where fatty acids and fatty alcohols coexist. I think it might be smart to check considering the mass balances.

The low mass balances at the very low temperature (180 °C) indicate that you have an issue with the determination of the lauric acid concentration. What usually happens with

saturated fatty acids is that their solubility at lower temperatures – after cooling down is low and they tend to precipitate, the trick is to sample them warm then dilute with a better solvent (we found iso-propanol to work best). This should improve the mass balance at lower temperatures in the future. For this manuscript I think the clarification that this is a possibility needs to be added.

The trend for the pressure variation data makes no sense and there is no kinetic model that could describe such behaviour (global minima between two pressures)⁷, which shows that the data is unreliable, and the results are likely guided by the mass balance results. I hope the authors can make that clear in the manuscript.

I do not understand why the reaction conditions were not optimized for the most active catalyst (PdNP/H-MZSM-DA) but rather for the PdNP/H-MZSM5). Additionally, most of the other characterization was done for the PdNP/H-MZSM5 and nearly none was done for the PdNP/H-MZSM5-DA. Please explain.

References

1. Cheng, K. et al. Maximizing noble metal utilization in solid catalysts by control of nanoparticle location. *Science* vol. 377 <https://www.science.org> (2022).
2. Kim, T. H., Lee, K., Kim, M. Y., Chang, Y. K. & Choi, M. Effects of Fatty Acid Compositions on Heavy Oligomer Formation and Catalyst Deactivation during Deoxygenation of Triglycerides. *ACS Sustain Chem Eng* 6, 17168–17177 (2018).
3. del Río, J. I., Pérez, W., Cardeño, F., Marín, J. & Rios, L. A. Pre-hydrogenation stage as a strategy to improve the continuous production of a diesel-like biofuel from palm oil. *Renew Energy* 168, 505–515 (2021).
4. Oh, M. et al. Importance of pore size and Lewis acidity of Pt/Al₂O₃ for mitigating mass transfer limitation and catalyst fouling in triglyceride deoxygenation. *Chemical Engineering Journal* 439, 135530 (2022).
5. Žula, M., Grilc, M. & Likozar, B. Hydrocracking, hydrogenation and hydro-deoxygenation of fatty acids, esters and glycerides: Mechanisms, kinetics and transport phenomena. *Chemical Engineering Journal* 444, (2022).
6. O'Brien, C. P. & Lee, I. C. CO Poisoning and CO Hydrogenation on the Surface of Pd Hydrogen Separation Membranes. *Journal of Physical Chemistry C* 121, 16864–16871 (2017).
7. Singh, U. K. & Vannice, M. A. Kinetics of liquid-phase hydrogenation reactions over supported metal catalysts-a review. *Applied Catalysis A: General* vol. 213 (2001).

Reviewer #4 (Remarks to the Author):

This paper presents a novel method for synthesizing doped Pd zeolites on H-ZSM-5. Remarkable control is shown over creating the active sites, specifically within micro and mesopores in the zeolite structure, to tune selectivity towards the desired HDO pathway. The authors provide a detailed overview of the novel synthesis method, and the zeolite is extremely well characterized. The article is well written. I recommend that this paper be published after the authors address the following comments.

Major Comments

1. Scheme 1 shows a cascade reaction of lauric acid to dodecane. The authors need to prove that this cascade is taking place by identifying the reaction intermediates or citing work that establishes this cascade.
2. The authors compare the production rate of the different zeolite catalysts after 120 mins. This was done at high conversions which could be misleading, it would be more effective to compare the production rates at lower conversions of lauric acid. A kinetic study here would be ideal. The same comment applies to the recyclability tests, they are usually conducted at low conversions of the reactant.
3. The authors claim that PdNP/H-MZSM5 exhibits good recyclability. However, from Supplementary Figure 14, it's clear that there is a decrease in the yield of dodecane. How many turnovers of lauric acid were observed over the 3 cycles?
4. The carbon balance of ~80% in the optimization studies is concerning, especially as the authors have shown that coking does not occur in the system. The reason or source of this carbon imbalance needs to be identified.
5. The authors claim that the species adsorbed onto the zeolite participate in the cascade process. There is no data to support this. This can be done by separating the spent zeolites and running a reaction without any additional reactant.
6. Fatty acids such as lauric are derived in a range of carbon chain lengths (C8-C18). Can the authors demonstrate the activity of the zeolite for other commonly found fatty acids?
7. Fatty acids can be relatively expensive to source compared with aviation fuels; recent work has shown that they can be upgraded to higher-value surfactants and lubricants through ketonization chemistry. Can the authors comment on the economics of this process?

Minor Comments

1. In Table.1 The authors need to report the turnover numbers for each catalyst screened
2. The reaction conditions i.e. temperature reactant loading need to be reported in figure 4 and the other supplementary figures where yield data is given.
3. Does the system have any external mass transfer limitations? Have the authors looked into testing the production rate at different stirring speeds?

Reviewer #2 (Remarks to the Author):

Ding et al. reported a bifunctional Pd/H-meso-ZSM-5 with Pd NPs selectively placed in mesopores, which exhibited superior performance in hydrodeoxygenation of lauric acid in comparison to the one with small Pd NPs in zeolite crystals. This finding is different from the general understanding of the closer the better. The size of Pd NPs and their location along mesoporous zeolite have well been characterized. Generally, this manuscript is well-written and organized. However, I cannot suggest its publication at this stage because of the following concerns.

(1) The findings of this work about the spatial organization of functional sites may benefit the development and production of multifunctional nanomaterials, but the performance improvement in HDO of lauric acid is not that significant compared with the literature (Catal. Sci. Technol., 2014, 4, 3705–3712), and even much lower. A comparison with published results should be made. Besides, based on the literature I looked through, it seems that the Pt-based catalysts are better than Pd-based catalysts and the price of Pt is even lower than that of Pd. The author should clarify why Pd was chosen for this material.

Response: We thank the reviewer for the positive but critical comment, which picks up on one of the driving factors behind the work, namely the design of spatial compartmentalisation of functional sites within a single catalytic material. This approach reflects our attempts to, at least partially, mimic natural systems through the development of spatial control within multifunctional materials, which we feel could benefit a plethora of catalyst processes. To demonstrate the true potential of such materials, they require deployment in multistep cascade processes in which distinct steps of the cascade occur at different catalytic sites, e.g. reduction over metal and dehydration over acid, for which the configuration employed enables. While the existing literature has identified Pd-doped alumina silicates as interesting hydrodeoxygenation (HDO) candidates (Microporous Mesoporous Mater., 2019, 280, 88-96, and Catal. Commun., 2012, 17, 76-80), the nature and potential interaction between metal and acid is unclear, not least given that the separation of the two types of catalytic site within separate species, i.e. a physical mixture of two materials (a mono functionalised metal catalysts and a solid acid zeolite) shows promise in HDO reactions (ACS Catal., 2017, 8, 785-789, and Green Chem., 2019, 21, 5046-5052). Thus, it leads us to hypothesise that while both sites are required, they do not need to be in direct contact with each other, a hypothesis that spatial compartmentalisation at the nanoscale, i.e. merely a few nm apart with our best-performing catalyst PdNP/H-ZSM-5-DA confirms, showing optimal performance over physical mixtures of two catalyst due to short diffusion distance between each site. Moreover, in the case of Pd systems, the absence of acidic sites results in only partial HDO, due to the capacity of Pd solely for the reduction of carboxylic acids to alcohols (Catal. Today, 2012, 185, 205-210, and ACS Catal., 2013, 3, 2327-2335), which is discussed in the introduction, i.e. setting out the cascade process. This is likewise observed when the zeolite from the physical mixture, outlined above (Green Chem., 2019, 21, 5046-5052), is removed from the system, i.e. the cascade stops at the fatty alcohol due to the removal of the dehydration catalytic active species. Moreover, the impact of relatively subtle differences in separation distance of active sites on catalytic performance has recently been reported (Nature Catal. 2024, 7, 172–184), and whilst the approach and reaction differ from the one we report, it likewise demonstrates that the perception that closer is better is not always the case.

On paper, the reviewer's suggestion to deploy Pt, especially if the sole goal was to produce the optimal performing system, would be valid. However, when comparing Pt vs Pd, albeit at

higher temperatures, Pt has a greater propensity for HDO than Pd, which can favour decarbonylation and decarboxylation (DCOx) (Fuel, 2012, 94, 578-585, and Mol. Catal. 2022, 523, 111492), i.e. the two metals behave differently providing different pathways for the conversion of fatty acids. It should be noted here that Pd, and not Pt, shows the best conversion performance but is let down by poor selectivity, which ideally would be overcome. Moreover, choosing Pt could, in fact, make the acid sites of the zeolite redundant, either particularly or fully, and therefore, not realise a multisite catalytic cascade process capable of fully demonstrating the capacity of spatial compartmentalisation, for which the use of Pd, and its ability to only partially HDO the acid to the alcohol does. Thus, based on the existing literature and our motivation to shed further insight into novel catalyst design and production, Pd and not Pt were deemed to be the preferred metal candidates. From this spatial separation of Pd and acidity within a single catalyst we have shed further insight into the level of interaction between these two specific sites required for fatty acid HDO, which indicates that for Pd-based systems, such as the Pd-doped alumina silicates (Microporous Mesoporous Mater., 2019, 280, 88-96, and Catal. Commun., 2012, 17, 76-80) the metal and acid site work separately within the overall cascade process, i.e. metal drives acid reduction to the alcohol which dehydrates over the acid sites. A second factor in driving this investigation was to expand on our previous work (Nat Mater, 2016, 15, 178-182), moving the direction towards the development of catalytic systems with industrial familiarity, i.e. zeolites, in conjunction with a facile synthesis, to increase industrial appeal. In regard to economics and the prices of Pd and Pt, recent JM data shows that the trade price of Pd is currently slightly lower than Pt (i.e., Pd average of \$979.82/troy oz vs \$1,031.14/troy oz for Pt), access date 08/06/2024. (<https://matthey.com/products-and-markets/pgms-and-circularity/pgm-management>).

A comparison of our system to the current literature is made in Supplementary Table 7, which initially focused on Pd systems and omitted solvent-free reactions, given the contribution solvent can impart on liquid phase kinetics and the benefit of higher substrate concentration on rate. However, we have further expanded this table to give a wider comparison to the current literature, including a comparison to the work reported in Catal. Sci. Technol., 2014, 4, 3705–3712. While this example from the literature would appear to show Pt to be preferential, we feel we should point out that the lack of characterisation of the different metal catalysts within the publication hinders a fair comparison, while cooling the catalyst under H₂ during the pretreatment could also play a role in the poorer performance of Pd due to its capacity of hydride formation (J. Phys. Chem. C 2009, 113, 15140–15147). Moreover, when evaluating mol normalised productivity, it is shown that our system performs comparably to the optimal Pt/Nb₂O₅ system reported by Kenichi et al.

Finally, we feel we should stress that the overarching aim of the manuscript is to demonstrate a new class of catalysts based on zeolites with the compartmentalisation of different active species and the benefit of such configuration within multistep multisite cascade processes rather than simply aiming to develop the most optimal catalyst.

Action: Manuscript and ESI amended.

Comparisons are also made with Pt based systems, with either comparable (Pt/Nb₂O₅) or superior (Pt/ZSM-5, Pt/TiO₂, Pt-Re/TiO₂) alkane productivities observed.

Supplementary Table 7. Comparison of different HDO catalysts.

Catalyst	Fatty acid to Pd mole ratio	Temperature and pressure	Reaction time	Conversion	HDO/DCO _x ^a	Productivity ^b (mol _{product} mol _{Pd} ⁻¹ h ⁻¹)	Ref
Pd/C	140.78	300 °C, 10 bar	5 h	96%	0.05	1.3 (27.0)	2
Pd/Al ₂ O ₃	29.86	350 °C, 14 bar	3 h	97%	0.1	0.8 (8.6)	2
Pd/Al-SBA-15	186.62	250 °C, 25 bar	3 h	65%	0.25	7.5 (37.6)	2
Pd/SAPO-31	9.54	320 °C, 20 bar	1 h	100%	0.25	1.6 (8.0)	2
Pd@Al-SiO ₂	23.79	260 °C, 30 bar	5 h	98%	2.5	3.3 (4.7)	3
Pd-Nb ₂ O ₅ /SiO ₂	10.43	170 °C, 25 bar	24 h	100%	20	0.4 (0.4)	4
Pd/HPA-SiO ₂	44.46	200 °C, 30 bar	3 h	100%	13	13.8 (14.8)	5
Pd/CuZnAl + ZSM5 ^c	113.37	200 °C, 20 bar	8 h	100%	100	14.0 (14.2)	6
Pd-Re/C + zeolite A	50.00	160 °C, 30 bar	6 h	87%	50	7.1 (7.3)	7
Pt/Nb ₂ O ₅ ^d	100	180 °C, 8 bar	4 h	100%	no DCO _x	22 (22)	8
Pt/ZSM-5	68.66	270 °C, 65 bar	12	100%	2.48 ^e	5.72	9
Pt/TiO ₂	81.67	130 °C, 20 bar	8	60%	10 ^f	6.13 (5.57)	10
Pt-Re/TiO ₂	81.67 ^g	130 °C, 20 bar	2.5	86%	7.1	9.36 (8.21) ^g	10
PdNP/H-MZSM5-DA	198.75	200 °C, 30 bar	6 h	94%	5.1	20.6 (24.6) ^h	This work

^a HDO/DCO_x was calculated based on (yield of HDO products)/(yield of DCO_x products). ^b Productivity based on HDO only, with productivity based on all products reported in parenthesis. In both cases, productivity is calculated based on the final conversion and selectivity reported for the corresponding reaction time, as reaction profiles are not provided in many of the literature reports. Where Pt has been used as the metal site productivity is based on these elements. ^c Pd acts to stabilise Cu, which is the proposed active species, rather than being the active species. However, given the higher cost and low earth abundance, productivity is calculated based on Pd and not Cu. ^d Solvent free conditions. ^e Based on the carboxylic acid. ^f Based on alcohol as an intermediate HDO product. ^g Calculated with only the alkane as the HDO product, i.e. full HDO. ^h Productivity calculated at 6 h for comparison with the existing literature, i.e. at the end of the reported reaction, whereas in Table 1 in the manuscript, it is determined for optimal performance at 2 h.

(2) *The dehydration of alcohol to alkene is kinetically feasible, not a rate-determining step. Why H-ZSM-5 with strong acid sites was employed. The investigation of the zeolite topology effect and support effect should be made to show the advantage of H-ZSM-5.*

Response: We thank the reviewer for the comment. To avoid potential confusion, we have amended the manuscript to remove the term rate-determining step and clarified the discussion around the rates of the different steps of the cascade.

As outlined above, the use of a strong acid, i.e. H-ZSM-5, is critical to the overall cascade process. Its removal has been shown to result in premature termination of the process with only partial HDO to the fatty alcohol (ACS Catal., 2017, 8, 785-789, and Green Chem., 2019, 21, 5046-5052). This is further demonstrated in entries 1-4 of Table 1. When Na-ZSM-5 is substituted in, the cascade slows and results in the formation solely of the fatty alcohol, consistent with the literature. Conversion of the fatty alcohol intermediate does exhibit a strong influence on fatty acid conversion, which we attribute to driving the equilibrium forward. As shown in Figure 4, the rate of alcohol dehydration is not impacted by the removal of acid sites within the mesopores, confirming that the micropores of H-ZSM-5 are essential for the dehydration process, with this step of the cascade occurring within them. Furthermore, the choice of zeolite framework employed was in part guided by the literature, in which ZSM-5 has been shown to allow full HDO when coupled with a Pd/CuZnAl catalyst.

The suggestion to investigate other zeolite frameworks is a logical evolution of the approach. We have investigated changing the Si:Al from 40 to 25 and 15, to increase the acid site loading. This showed no benefit to the cascade process, which is not surprising given the significantly higher kinetic of the dehydration step vs the reduction step, but it does demonstrate the flexibility of the approach. This tunability can also be applied to the choice of framework, with the successful expansion of the catalyst synthesis protocol to H-beta and USY zeolites, with catalytic data reported in Supplementary Table 6.

Action: Manuscript and ESI amended.

The initial reduction step of the acid to the alcohol is, therefore, slower than the subsequent alcohol dehydration step, which is further supported given that cascade intermediate species are not observed, i.e., intermediates are consumed rapidly within the pore architecture of the bifunctional catalysts as substrates for the subsequent step in the cascade before having time to diffuse into the bulk reaction media. Likewise, alkene hydrogenation is also a facile step in the cascade and is thus consumed before building up in the reaction media.^{36,37}

To further explore this approach and evaluate the impact of intrinsic zeolite acidity, the optimal synthetic route was applied to ZSM-5 with Si:Al ratios of 25 and 15. Increasing the inherent acidity of the catalytic system shows little effect on overall conversion (Supplementary Table 6), note the higher substrate concentration and lower metal loading, with only a decrease observed at the lowest Si:Al ratio. In contrast, selectivity towards HDO is compromised by the increasing acidity, which negatively impacts C₁₂ alkane productivity. Furthermore, switching

to other zeolite frameworks, namely USY and Beta, further demonstrates the generic nature of the approach for catalyst synthesis. While resulting in superior acid conversion, it is at the expense of C₁₂ alkane selectivity.

Table 1 | Lauric acid HDO performance over conventional prepared and spatial compartmentalised catalysts.^a

Catalysts	Acid Conv., %	C ₁₂ Sel., %	C ₁₁ Sel., %	Productivity ^b (mol _{C₁₂} mol _{Pd} ⁻¹ h ⁻¹)	TON ^c
H-MZSM5	0	0	0	0	0
PdNP/Na-MZSM5	2.2	27.3 (alcohol)	0 ^d	0.05 (alcohol)	29
H-MZSM5 + PdNP/Na-MZSM5 ^e	20.6	28.1	2.1	1.7	278
PdNP/H-MZSM5	51.0	51.8	9.9	11.9	689
Pd _{imp} /H-MZSM5	31.2	38.4	20.5	3.6	187
PdNP/ H-MZSM5-DA	94.1	64.9	12.6	30.6	1272
Pd _{imp} / H-MZSM5-DA	70.9	39.9	18.2	11.3	426
PdNP _{small} /H- MZSM5	50.9	40.4	14.9	9.9	688
PdNP _{small} /H- MZSM5-DA ^f	71.5	51.9	20.3	27.0	1205

^a Reaction conditions: 100 mg 1 wt.% Pd-doped catalyst, 300 mg lauric acid, 0.1 cm³ of nonane as the internal standard, 40 cm³ of hexane as the solvent, at 200 °C, 30 bar and 1000 rpm for 6 h. ^b Production rate was calculated based on the yield of n-C₁₂ + iso-C₁₂ in 2 hours, as shown in Fig. 4 (a). The unit is mol_{dodecane} mol_{Pd}⁻¹ h⁻¹, in which the quantity of Pd is based on Pd loading from ICP. ^c TON calculated based on moles of lauric acid converted at 6 h by surface Pd. ^d The concentration of C₁₁ is below the GC detection limit. ^e Physical mixture of 100 mg H-MZSM5 and 100 mg PdNP/Na-MZSM5. ^f 80 mg 1 wt.% Pd-doped catalyst. The errors of conversion and selectivity are within ±3%, whereas yields and production rates are within ±5%.

Supplementary Table 6. Lauric acid HDO performance over conventional prepared and spatial compartmentalised catalysts.^a

Catalysts	Acid Conv., %	C ₁₂ Sel., %	C ₁₁ Sel., %	Production rate ^b
PdNP/ H-MZSM5-DA	66.7	57.4	12.5	71.0
PdNP/ H-MZSM5-DA25	73.0	50.5	9.6	53.3
PdNP/ H-MZSM5-DA15	51.7	27.9	5.6	23.3
PdNP/H-USY-DA	99.3	52.7	24.0	139.1
PdNP/H-BETA-DA	83.3	47.4	19.6	93.7

^a Reaction conditions: 100 mg 0.4 wt.% Pd-doped catalyst, 600 mg lauric acid, 0.1 cm³ of nonane as the internal standard, 40 cm³ of hexane as the solvent, at 200 °C, 30 bar and 1000 rpm for 6 h. ^b Production rate was calculated based on the yield of n-C₁₂ + iso-C₁₂ in 2 hours, as shown in Fig. 4 (a). The unit is mol_{dodecane} mol_{Pd}⁻¹ h⁻¹, in which

the quantity of Pd is based on Pd loading from ICP. The errors of conversion and selectivity are within $\pm 3\%$, whereas yields and production rates are within $\pm 5\%$.

(3) *The Pd size of the optimum PdNP/H-MZSM5 catalyst is about 4.4 nm with a Pd dispersion of 26.5%, which means the utilization of Pd is not very high. The particle size effect or structure sensitivity of Pd NPs should be investigated. At least, PdNP_{small}/H-MZSM5-DA should be included.*

Response: The reviewer makes a good point, one we thoroughly concur with, which could be improved on. However, as outlined above, the aim of the initial investigation was the development of spatial control with a system of commercial and industrial relevance. To clearly demonstrate this, we have targeted Pd NPs that are larger than the micropores of the zeolite to ensure size control over their location. This included ensuring that the distribution of Pd nanoparticles (NPs) does not contain Pd NPs of less than 1 nm, i.e. larger than the micropores of ZSM-5 (0.74 nm). Having confirmed the spatial control in PdNP/H-ZSM-5 and PdNP/H-ZSM-5-DA, using Pd NPs of 4.5 nm, we are able to turn our attention to more efficient and sustainable use of Pd through decreasing Pd NP size, as reported for PdNP_{small}/H-ZSM-5. Decreasing Pd NP size for the system not subjected to mesopore dealumination shows no benefit to the catalytic cascade, with smaller Pd NPs increasing DCO_x at the expense of HDO, suggesting that Pd NP particle size affects the cascade pathway, consistent with the literature which shows an inverse correlation between Pd particle size and decarbonylation activity (ChemSusChem, 2016, 9, 3441-3447). To further demonstrate this, we have synthesised and screened PdNP_{small}/H-MZSM5-DA, which is consistent with the previous results and the discussion here, i.e. increased surface Pd sites, via smaller Pd nanoparticles, does not induce greater performance with activity comparable to large NP on equivalent supports while simultaneously negatively impacting on HDO selectivity (via increased C₁₁ selectivity). The positive impact observed for PdNP_{small}/H-MZSM5-DA is once again the result of realising true spatial compartmentalisation of active species. It is thus further evidence of the proposed multistep cascade which occurs as discrete steps over different active species residing in different pore networks of the hierarchical zeolite.

Action: Manuscript and ESI amended.

As previously observed in the non-dealumiated system, the incorporation of small Pd NPs (PdNP_{small}/H-MZSM5-DA) with a greater number of surface sites is not conducive to improved activity relative to its larger NP equivalent (PdNP/H-MZSM5-DA). Likewise, it induces increasing C₁₁ formation, although now solely due to reduced NP size.⁶⁹ That said, achieving true compartmentalisation once again is highly beneficial, as evident by the amplified activity and conversion relative to the impregnated and non-dealuminated equivalents (Pd_{imp}/H-MZSM5-DA and PdNP_{small}/H-MZSM5, respectively). While the former does give rise to a significant increase in performance relative to Pd_{imp}/H-MZSM5, it does not reach the level of PdNP/H-MZSM5-DA or PdNP_{small}/H-MZSM5-DA, confirming the benefit of spatial orthogonality of active species on catalytic HDO.

Table 1 | Lauric acid HDO performance over conventional prepared and spatial compartmentalised catalysts.^a

Catalysts	Acid Conv., %	C ₁₂ Sel., %	C ₁₁ Sel., %	Productivity ^b (mol _{C12} mol _{Pd} ⁻¹ h ⁻¹)	TON ^c
H-MZSM5	0	0	0	0	0
PdNP/Na-MZSM5	2.2	27.3 (alcohol)	0 ^d	0.05 (alcohol)	29
H-MZSM5 + PdNP/Na-MZSM5 ^e	20.6	28.1	2.1	1.7	278
PdNP/H-MZSM5	51.0	51.8	9.9	11.9	689
Pd _{imp} /H-MZSM5	31.2	38.4	20.5	3.6	187
PdNP/ H-MZSM5-DA	94.1	64.9	12.6	30.6	1272
Pd _{imp} / H-MZSM5-DA	70.9	39.9	18.2	11.3	426
PdNP _{small} /H- MZSM5	50.9	40.4	14.9	9.9	688
PdNP _{small} /H- MZSM5-DA ^f	71.5	59.3	24.4	28.1	1205

^a Reaction conditions: 100 mg 1 wt.% Pd-doped catalyst, 300 mg lauric acid, 0.1 cm³ of nonane as the internal standard, 40 cm³ of hexane as the solvent, at 200 °C, 30 bar and 1000 rpm for 6 h. ^b Production rate was calculated based on the yield of n-C₁₂ + iso-C₁₂ in 2 hours, as shown in Fig. 4 (a). The unit is mol_{dodecane} mol_{Pd}⁻¹ h⁻¹, in which the quantity of Pd is based on Pd loading from ICP. ^c TON calculated based on moles of lauric acid converted at 6 h by surface Pd. ^d The concentration of C₁₁ is below the GC detection limit. ^e Physical mixture of 100 mg H-MZSM5 and 100 mg PdNP/Na-MZSM5. ^f 80 mg 1 wt.% Pd-doped catalyst. The errors of conversion and selectivity are within ±3%, whereas yields and production rates are within ±5%.

(4) *The authors stated that mesopore hydrophobicity aids the expulsion of water (the by-product of acid reduction), thus increasing alcohol production. However, if this is true, the hydrophobicity will also suppress the diffusion of lauric acid to Pd NPs, which are deposited on hydrophobic surfaces.*

Response: The reviewer highlights an interesting point regarding the impact of hydrophobicity of the zeolite framework. While increasing hydrophobicity would be expected to impact on the interaction/diffusion of the polar head group of lauric acid, we suspect this is overshadowed (at least partially) by the presence of the C₁₂ hydrocarbon chain, i.e. the fatty component, which would govern significantly the average interaction of the substrate and the zeolite surface within the mesopores. To elucidate whether this is the case, the interaction of 1-nonanoic acid (the longest C chain room temperature liquid carboxylic acid) and 1-nonanol with H-ZSM-5 and H-ZSM-5DA has been investigated. The results confirm that the interaction of these species with the two zeolites is an intermediate between the two original probes (water and octane), i.e. the long hydrocarbon chain of the carboxylic acid (and alcohol) does indeed have a significant impact on the interaction with the catalyst, with the effect, albeit slightly, more greatly impacting the carboxylic acid. More interestingly, though, the influence of dealumination, i.e. increasing hydrophobicity, shifts the nature of the interaction of both compounds towards that of the hydrocarbon probe, and thus, increasing hydrophobicity does not negatively impact carboxylic acid interaction with the support. This being consistent with

the initial NMR measurements reported, which show weaker and stronger interactions of the dealuminated system with water and n-octane, respectively. Moreover, the rates of acid conversion (Table 1 and Figure 4) over the first 1 h are significantly higher for the dealuminated system, inconsistent with suppressed diffusion of lauric acid. This is further supported by literature, as outlined in the manuscript, for which it has been reported that increasing support (including zeolites) hydrophobicity increases fatty acid esterification rates, for which water is the by-product (Chem. Soc. Rev., 2014, 43, 7887-7916). Thus, increased support hydrophobicity both here and in the literature appears not to hinder the diffusion of long chain fatty acids.

Action: Manuscript and ESI amended.

The interaction of a model substrate and intermediate, 1-nonanoic acid and 1-nonanol, respectively, with the zeolites (H-MZSM5 and H-MZSM5-DA) can be considered to arise from a combination of the non-polar hydrocarbon chain and polar functional group, i.e. an intermediate of the two initial probes (water and octane) rather than one dominating over the other (shown in Supplementary Fig. 8 and Supplementary Table 3). Dealumination results in an increase in interaction for both model substrate and intermediate, reflecting a significant affinity between the hydrocarbon chain and the increased hydrophobic mesopore surface rather than a decreased interaction due to the repulsion of the polar functional group.

Concurrently, increasing hydrophobicity within PdNP/H-MZSM5-DA does not negatively impact interaction with the fatty acid (discussed above) and, thus, would not negatively impact cascade process efficiency.

Supplementary Figure 8. T_1 and T_2 relaxation times plots of 1-nonanoic acid (a, b, e, f) and 1-nonanol (c, d, g, h) within H-MZSM5 (a–d) and H-MZSM5-DA (e–h) catalysts.

Supplementary Table 3. T_1 , T_2 and T_1/T_2 values of different probes imbibed within the pores of the hierarchical ZSM-5.

H-MZSM5

H-MZSM5-DA

Responses to reviewers' comments: Ms. No. NCOMMS-23-43524-T

Chemical	T_1 (ms)	T_2 (ms)	T_1/T_2 (-)	T_1 (ms)	T_2 (ms)	T_1/T_2 (-)
Water	16 ± 0.8	9 ± 0.5	1.8 ± 0.1	24 ± 1.2	16 ± 0.8	1.5 ± 0.1
1-nonanol	126 ± 6.3	38 ± 1.9	3.3 ± 0.2	104 ± 5.2	29 ± 1.5	3.5 ± 0.2
1-nonanoic acid	150 ± 7.5	37 ± 1.9	4.1 ± 0.2	170 ± 8.5	36 ± 1.8	4.7 ± 0.2
Octane	117 ± 5.8	11 ± 0.6	10.6 ± 0.5	185 ± 9.2	12 ± 0.6	15.4 ± 0.8

(5) The title is too big based on the results in the manuscript. The authors only used lauric acid as a probe molecular for fatty acids. Some other fatty acids with different carbon numbers should be performed.

Response: The reviewer makes a fair comment and given the importance of expanding this study to demonstrate the advancement realised from this work and the application of this catalyst, we have expanded the study to include palmitic (C16) and stearic (C18) acids, both common compounds found in plant oils.

Action: Manuscript and ESI amended.

The inherent benefit of the unique active site spatial compartmentalisation within PdNP/H-MZSM5-DA is further evident from its capacity for HDO of other fatty acids, including palmitic (C16) and stearic (C18) (Supplementary Table 5).

Supplementary Table 5. PdNP/H-MZSM5-DA performance for HDO of fatty acids.^a

Substrate	Acid Conv., %	C _n Sel., %	C _{n-1} Sel., %	Production rate ^b
Palmitic acid	19.2	52.4	14	13.6
Stearic acid	17.5	78.4	21.6	12.5

^a Reaction conditions: 34 mg 1 wt.% Pd-doped catalyst, 380 mg palmitic or 420 mg stearic acid, 0.1 cm³ of nonane as the internal standard, 40 cm³ of hexane as the solvent, at 200 °C, 30 bar and 1000 rpm for 6 h. ^b Production rate was calculated based on the yield of C_n in 2 h. The errors of conversion and selectivity are within ±3%, whereas yields and production rates are within ±5%.

Reviewer #3 (Remarks to the Author):

Manuscript review comments

Review on the article: "Spatial segregation of catalytic sites within Pd doped H-ZSM-5 for fatty acids hydrodeoxygenation to alkanes". The authors have achieved commendable activity through accessibility of Pd active sites. Beyond that the article is well written and offers some excellent microscopy results and a very interesting approach to determination of the acid sites

accessibility and strength. However, I think there is a fundamental issue with this catalyst design idea. The separation of acid and metal sites in such setup is detrimental to catalyst stability. The way this setup differs from the hydroisomerization where the approach works¹ is that the acidic sites here can form their own alkenes via the dehydration of the alcohols which enables very high potential for all the reactions that are catalysed from the protonation of the double bonds – oligomerization, coking and so on. Additionally, these reactions have a concentration (second order or higher) and temperature dependence therefore the issue would increase exponentially with the reaction feed concentration increase and with temperature increase – which are both crucial for the consideration of the industrial use. The issue of acidity and high degree of unsaturation was addressed a few times in the literature including 2,3. The stability and the combination of acidic and hydrogenation function in proximity are the reason why transition metal sulphides (NiMoS / CoMoS) are used in industrial applications 4. Furthermore, I believe the superior activity is the result of the combination increase in the pore size⁵, Pd nanoparticles size and increased HDO selectivity which in turn prevents “CO poisoning” 6 and reactions such as methanation due to better dehydration activity of the catalyst – the acid sites not being blocked by the Pd deposition in the Pd/MZSM5-DA case. Based on these reasons, my conclusions on the presented data would be that the sites still work together while not blocking each other which happens with the impregnation methods and that the key to their activity is in accessibility rather than separation and the data presented has not convinced me otherwise. Notably, there is no data on the acidity of the metal deposited catalysts which may vary significantly and guide the activity of the catalyst. Regardless of that, I believe that article should be published after the issue is described throughout the manuscript and other corrections are applied due to high activity, an interesting approach and overall good presentation of the results.

Response: We thank the review for their alternative views on the process within our systems and their insight. The review suggests that the spatial separation reported within our system could be detrimental to catalyst stability, in particular, suffering from potential oligomerisation and coking. TGA of the spent catalyst, as shown in Supplementary Fig. 17, does not reveal an obvious mass loss at the temperature required for coke combustions, i.e. > 400 °C. Thus, under the “low” reaction temperatures employed, we suppress coke formation. The mass loss at less than 300C is attributed to the desorption of adsorbed substrate, intermediates, and products for the cascade process. We propose that under the reaction conditions studied and from the unique spatial geometry of the two classes of active species within the zeolite framework, rapid hydrogenation of the alkene to the alkane occurs, over the Pd NP within the mesopores, i.e. as the alkene diffuses out from the catalyst micropore framework (as discussed in the manuscript). Thus, the possibility of coking/oligomerization is suppressed. The oligomerisation observed in the reported literature results from these being investigations into unsaturated triglycerides and the potential for oligomerisation of these due to their unsaturation. However, here we have generally focused on a saturated fatty alcohol, which could explain the variation in unwanted side reactions from our study and the literature. We have investigated increasing substrate concentration, and based on comparable C12 selectivity when employing double the substrate concentration (Supplementary Table 6), we are still confident under the lower temperature studied, exacerbated coking/oligomerisation is not an issue. While this may be a factor at higher temperatures, part of the aim of this work is to develop systems with high activity at mild conditions, which obviously would be beneficial from an energy perspective and thus desirable industrially. If pore size and Pd particle size were dictating factors the performance of PdNP/H-ZSM-5 should be superior (initial activity),

given that the Pd is comparable in size but the pores (being on external surfaces only) would be infinitely larger while also overcoming the issue of Pd blocking acid site within the framework. However, this is not the case. We agree that the high HDO selectivity is critical in preventing undesirable CO poisoning of the Pd sites and that their spatial separation from them also prevents acid site blocking (again discussed in the manuscript), which is an additional benefit of our approach. However, we do point out that acid sites are present in the parent zeolite (H-MZSM-5) in a 7.6-fold excess to the number of Pd atoms impregnated. Furthermore, as Pd forms NPs, only 26.5% of the Pd atoms would be available (present at the surface) for acid site blocking, resulting in a maximum of only ~ 3.5% of acid site being blocked. An insignificant proportion, given the significant impact on catalytic performance. Regarding Pd size and potential negative impact of smaller Pd within the impregnated, while smaller Pd does favour decarbonylation, it appears, based on the comparison of PdNP_{small}/H-MZSM5 and Pd_{imp}/H-MZSM5 which possess comparable sized Pd NPs (Average size ~ 2.5 nm), not to be the sole factor, with PdNP_{small}/H-MZSM5 resulting in a 25% decrease in decarbonylation and twice the yield of dodecane. This suggests that spatially segregating Pd within the more accessible mesopore, directly accessible from the external surface of the zeolite, rather than throughout the framework, is beneficial, with potential from more facile solvent and H₂ diffusion to them and reduced interaction between acid and Pd sites. Depositing small Pd on H-MZSM5-DA (PdNP_{small}/H-MZSM5-DA) further confirms small Pd is not beneficial, i.e. lower selectivity than PdNP/H-MZSM5-DA, but does show superior C₁₂ selectivity to PdNP_{small}/H-MZSM5, Pd_{imp}/H-MZSM5 and Pd_{imp}/H-MZSM5-DA, and thus further confirms size is not the sole factor, and that spatial segregating both sites is beneficial to HDO. Regarding the final point on the omission of acidity measurements on the metal deposited catalysts, such measurements are complicated by the potential for ammonia adsorption on Pd, and its desorption over the range of 508-603 K (J. Phys. Chem. C, 1986, 90, 2906-2910), which partly overlaps with its desorption from the acid sites of the zeolite.

Action: Manuscript amended where appropriate.

We propose that the spatial segregation within the reported catalytic systems should be beneficial to a cascade hydrodeoxygenation that proceeds stepwise via reduction (carboxylic acid to alcohol) in the mesopore over Pd, dehydration (alcohol to alkene) in the micropores over acid sites, and finally hydrogenation (alkene to alkane) over Pd in the mesopores, with the alkane product then diffusing to the bulk reaction media. I.e. cascade intermediates are contained within the zeolite framework and subsequently consumed until the final product is generated. The stepwise cascade pathway is illustrated in Scheme 1, which is also observed for Ni/ZSM-5 and Mo/ZSM-22^{64,65}

This varies from studies into unsaturated triglycerides where coking and oligomerisation are apparent due to the reactivity of the unsaturation in the initial substrate⁷¹.

Moreover, this approach prohibits Pd blocking/binding to the acid sites, another undesirable aspect of producing such bifunctional material. However, we do point out that here, within our Pd_{imp}/H-MZSM5, the acid site of the parent zeolite significantly outweighs the amount of Pd

impregnated, and thus, the resulting surface Pd atoms could, in theory, only block a maximum of 3.5% of the acid sites present, i.e. the impact of such an effect can be considered inconsequential.

Comments

“Hierarchical mesoporous-microporous ZSM-5 zeolite is deployed as both catalysts, given its inherent acidity, and catalyst support to host deposited metal sites.” – As this is indeed true, it is unusual that related previous studies are not that well reviewed, as there are quite some existing.

Aspects to thus be considered:

- the hydrocracking, hydrogenation and hydro-deoxygenation of bio-based fatty acids, esters and glycerides, considering mechanisms, kinetics and transport phenomena resistances*
- the conversion of the bio-based fatty acids over bi-functional Ni/ZSM-5 catalysts, considering the effect of stoichiometric Ni/Al molar ratio*
- the mechanism, ab initio calculations and micro-kinetics of the straight chain alcohol, ether, aldehyde, bio-based carboxylic acid and ester hydro-deoxygenation reactions over Mo–Ni catalysts*

Response: We thank the review for their suggestion and insight and have enhanced the introduction by the inclusion of a summary outlining the points most associated with the work reported within the manuscript based on recent literature (Chem. Eng. J. 2019, 359, 1339-1351, Chem. Eng. J. 2022, 444, 136564, and Top. Catal, 2018, 61, 1757-1768)

Action: Manuscript amended.

The reaction pathway, based on experimental and kinetic modelling of biomass derived species, is considered a multistep cascade process. Carboxylic acids are sequentially reduced to the corresponding alcohol, via the aldehyde intermediate, over metal sites. Subsequent alcohol dehydration yields alkenes, which in turn can undergo facile hydrogenation to saturated hydrocarbons while retaining the original carbon chain length. The alcohol intermediate may also undergo direct hydrogenation to the alkane, attributed to the absence of alkene detection although this may reflect the significant low activation energy of C=C hydrogenation, while the formation of esters, ethers and aldol condensation products are other potential side reactions^{36,37}. These latter pathways provide a circular route via their conversion back into the constituent species. Decreased carbon chain length species result from decarboxylation and decarbonylation, with the intermediate aldehyde a potential candidate for the latter, for which Ni/ZSM-5 has shown high and tunable selectivity towards^{37,38}. Overall, catalyst composition governs the propensity towards this array of different reactions (e.g. HDO, DCO_x, coupling), with choice of metal, acidity (including that of the support) and mass transportation, either singularly or combined, controlling factors^{37,38}.

The reaction scheme (Scheme 1) lacks a hydrogen molecule, two H₂ molecules are needed to go from fatty acid to alcohol, usually the hydrogenation of fatty aldehydes is very fast over noble metals thus those can not be seen on GC-MS.

Response: We thank the reviewer for highlighting this error in scheme 1.

Action: Manuscript amended.

Scheme 1. HDO reaction pathway of lauric acid to dodecane.

The supplementary Fig. 11 (now Supplementary Fig. 12), applying a longer analysis method might unravel some esterification product in the form of lauryl laurate. It's a common product of weaker acid sites, especially at lower temperatures in the reactions where fatty acids and fatty alcohols coexist. I think it might be smart to check considering the mass balances.

Response: The reviewer highlights an interesting point regarding the possibility of esterification between the alcohol intermediate and the acid. We do observe a trace amount of lauryl laurate (using the existing analytical method), over the PdNP/H-MZSM5-DA catalyst. However, its concentration does not increase significantly over the duration of the reaction, and thus, it would appear to be consumed. It was omitted previously not over to complicate the message of the manuscript; however, we can see merit in its inclusion and respect the reviewer's insight and recommendation.

Action: Manuscript amended.

The process also yields undecane (C₁₁), through undesirable DCO_x, this being the main by-product detected (Supplementary Fig. 12), while trace amounts of lauryl laurate from the esterification reaction between the fatty alcohol and acid are also detected. The concentration of the latter does not increase significantly over the duration of the reaction, suggesting its formation is a minor pathway.

The low mass balances at the very low temperature (180 °C) indicate that you have an issue with the determination of the lauric acid concentration. What usually happens with saturated fatty acids is that their solubility at lower temperatures – after cooling down is low and they tend to precipitate, the trick is to sample them warm then dilute with a better solvent (we found iso-propanol to work best). This should improve the mass balance at lower temperatures in the future. For this manuscript I think the clarification that this is a possibility needs to be added.

Response: We thank the reviewer for their insights. We sample the reaction at temperature and thus feel that, at least to a degree, the missing mass is due to some adsorption on the catalysts. However, some precipitation within the sampling system may be occurring. In

between samples, the system is flushed between samples to overcome the possibility of carryover.

Action: Manuscript amended.

It is important to note that due to the potential for precipitation of the fatty acid substrate at room temperature, sampling should be conducted at reaction temperature, and the sampling system should be flushed prior to sample collection to avoid carryover. Loss of the substrate during sampling will impact solely on the conversion reported, there are no issues with regard to product solubility.

The trend for the pressure variation data makes no sense and there is no kinetic model that could describe such behaviour (global minima between two pressures)⁷, which shows that the data is unreliable, and the results are likely guided by the mass balance results. I hope the authors can make that clear in the manuscript.

Response: We agree with the review comments regarding a minima between two pressures. However, the difference between 10 and 20 bar are within the error of the analysis, i.e. conversions are comparable (and mass balance may also be a slight factor), while dodecane yields increase with pressure. The by-product undecane formation results from two indistinguishable pathways, i.e. decarboxylation and decarbonylation, which do not require hydrogen and can even be suppressed by it. Hence, the high undecane formation at 10 bar, while yields at 20 and 30 bar are within experimental error. The errors in conversion and yields are reported in the figure captions.

Action: Manuscript amended

The impact of pressure shows at ≤ 20 bar only selectivity is influenced whilst conversion is comparable, whereas increasing pressure to 30 bar drives up conversion while maintaining higher C₁₂ selectivity. The impact of stirring rate shows no effect, as presented in Supplementary Fig. 11, ruling out bulk mass diffusion effects.

I do not understand why the reaction conditions were not optimized for the most active catalyst (PdNP/H-MZSM-DA) but rather for the PdNP/H-MZSM5). Additionally, most of the other characterization was done for the PdNP/H-MZSM5 and nearly none was done for the PdNP/H-MZSM5-DA. Please explain.

Response: Ideally, we would have employed the optimal catalyst for both the process condition optimisation and extensive catalyst characterisation. However, the optimal catalyst (PdNP/H-MZSM5-DA) only originated from the further development of the initial catalyst (PdNP/H-MZSM5). This was after initial studies, which included both reaction condition optimisation and electron tomography, i.e. PdNP/H-MZSM5 was the starting point of the investigation. The optimisation (further development) arose from the desire to completely rule out the potential for interaction between acid sites in the mesopore and the Pd NPs. Repeating

these studies (process optimisation and tomography) seemed excessive, given the similarities between the systems, especially with regard to characterisation techniques such as electron microscopy/tomography (which is time and cost demanding). Moreover, given the additional step in the optimal catalyst synthesis and reduced scale of its production (note, we anticipate scale-up would be facile but was not the focus here), process optimisation would have required multiple catalyst syntheses for all reaction conditions and for detailed characterisation. Hence, it was preferable to conduct optimisation on a single batch of catalyst, which was possible for PdNP/H-MZSM5.

Regarding other characterisation techniques applied to the original catalyst and not to the optimised one, these include N₂ porosimetry and HAXPES. The former showed no significant impact from the impregnation of preformed Pd NP onto H-MZSM5, which is highly likely to be the case in the dealuminated system, given the comparable porosimetry of the parent zeolite and it post dealumination. The latter was used to support Pd location evaluation based on Pd:Si semi-bulk analysis and its comparison to XPS and elemental analysis. It is reasonable to assume it will be comparable between PdNP/H-MZSM5-DA and PdNP/H-MZSM5, given the synthesis approach and subtle differences between H-MZSM5 and H-MZSM5-DA. To confirm or discredit whether these assumptions were correct, we have conducted HAXPES and N₂ porosimetry during the revision. Nitrogen porosimetry shows that, as with the non-dealuminate system, PdNP impregnation of the dealuminated support has no significant effect on the porous nature of the catalyst while also confirming the comparable porosity of PdNP/H-MZSM5-DA and PdNP/H-MZSM5, while surface, semi-bulk, and bulk Pd:Si, from XPS, HAXPES, and elemental analysis, respectively, show excellent agreement between PdNP/H-MZSM5-DA and PdNP/H-MZSM5. The extra data and corresponding discussion have been added.

Action: Manuscript and ESI amended

Supplementary Fig. 9 and Supplementary Table 1 reveal its analogous nature to its parent zeolite (where applicable) and PdNP/H-MZSM-5, with equivalent porosity (surface areas, pore volumes and pore diameter), Pd NP size (4.5 ± 0.9 nm from HAADF-STEM), and Pd:Si atomic ratios from surface, semi-bulk, and bulk analysis (0.0125 for XPS, 0.0118 for HAXPES, and 0.0048 for bulk analysis, to be compared with Fig.1c) substantiating their comparable physicochemical nature

Supplementary Table 1. Porous properties of the ZSM-5 zeolites and Pd-doped zeolite catalysts.

Sample	S_{BET} (m ² g ⁻¹)	S_{micro} (m ² g ⁻¹)	$S_{external}$ (m ² g ⁻¹) ^a	V_{micro} (cm ³ g ⁻¹) ^b	V_{meso} (cm ³ g ⁻¹) ^c	PD (nm) _d
HZSM-5	344 ± 34	208 ± 21	136 ± 14	0.109 ± 0.01	0.061 ± 0.006	<2
H-MZSM5	419 ± 42	206 ± 21	213 ± 21	0.108 ± 0.01	0.317 ± 0.032	8.6

Responses to reviewers' comments: Ms. No. NCOMMS-23-43524-T

H-MZSM5-DA	434 ± 43	177 ± 18	257 ± 26	0.093 ± 0.01	0.342 ± 0.034	8.9
Na-MZSM5	345 ± 34	222 ± 22	123 ± 12	0.116 ± 0.01	0.304 ± 0.030	6.9
Pd _{imp} /H-MZSM5	414 ± 41	193 ± 19	221 ± 22	0.101 ± 0.01	0.328 ± 0.033	8.7
PdNP/H-MZSM5	429 ± 43	205 ± 20	224 ± 22	0.108 ± 0.01	0.298 ± 0.030	8.4
PdNP/H-MZSM5- DA	441 ± 44	226 ± 23	215 ± 20	0.130 ± 0.01	0.321 ± 0.032	8.0

^a External surface area was determined by $S_{external} = S_{BET} - S_{micro}$; ^b determined by the *t*-plot method; ^c mesopore volume was determined by $V_{meso} = V_{total} - V_{micro}$; ^d PD = pore diameter, calculated from the adsorption branch using the BJH method with Fass correction.

Reviewer #4 (Remarks to the Author):

This paper presents a novel method for synthesizing doped Pd zeolites on H-ZSM-5. Remarkable control is shown over creating the active sites, specifically within micro and mesopores in the zeolite structure, to tune selectivity towards the desired HDO pathway. The authors provide a detailed overview of the novel synthesis method, and the zeolite is extremely well characterized. The article is well written. I recommend that this paper be published after the authors address the following comments.

Major Comments

1. Scheme 1 shows a cascade reaction of lauric acid to dodecane. The authors need to prove that this cascade is taking place by identifying the reaction intermediates or citing work that establishes this cascade.

Response: We have provided existing literature in support of the proposed cascade (Appl. Catal. A: Gen., 2014, 471, 28-38. Green Chem., 2016, 18, 4633-4648.) and clarified the overall process in more detail, including a revised paragraph in the introduction.

Action: Manuscript amended

We propose that the spatial segregation within the reported catalytic systems should be beneficial to a cascade hydrodeoxygenation that proceeds stepwise via reduction (carboxylic acid to alcohol) in the mesopore over Pd, dehydration (alcohol to alkene) in the micropores over acid sites, and finally hydrogenation (alkene to alkane) over Pd in the mesopores, with the alkane product then diffusing to the bulk reaction media. I.e. cascade intermediates are contained within the zeolite framework and subsequently consumed until the final product is

generated. The stepwise cascade pathway is illustrated in Scheme 1, which is also observed for Ni/ZSM-5 and Mo/ZSM-22^{64,65}. Observing cascade intermediate species is dependent on the catalyst configuration and species reactivity, e.g. carbonyl and alkene intermediates are often not observed due to rapid conversion^{36,37}, i.e. consumed before accumulating in the bulk solution.

2. The authors compare the production rate of the different zeolite catalysts after 120 mins. This was done at high conversions which could be misleading, it would be more effective to compare the production rates at lower conversions of lauric acid. A kinetic study here would be ideal. The same comment applies to the recyclability tests, they are usually conducted at low conversions of the reactant.

Response: The review is indeed correct with regard to conducting kinetic and recycle tests at low conversion, a common issue within the field. We do point out, as per the reaction profiles in Supplementary Figure 14, that the conversion of lauric acid was less than 35% for most catalysts studied and had reached only ~70% for PdNP/H-MZSM5-DA. While this would be an issue if we were to make a comparison based on conversion, it would not when basing it on C12 alkane yield, i.e. the product from the complete cascade. From Fig. 4, a reasonable linear increase in dodecane yield is apparent for all systems over the initial 2 hours of the reaction (after an initial induction period), and thus, changing the calculation to the initial 1 hour would not significantly impact the productivities reported (for example PdNP/H-MZSM5-DA would increase to 30.6 would increase only to 33.7). The recycle studies were conducted at ~40% conversion for PdNP/H-MZSM5 and PdNP/H-ZSM5, and ~65% conversion for PdNP/H-MZSM5-DA, significantly below the recommendation of avoiding studies at close to full conversion.

Action: None

3. The authors claim that PdNP/H-MZSM5 exhibits good recyclability. However, from Supplementary Figure 14 (now Supplementary Fig. 14), it's clear that there is a decrease in the yield of dodecane. How many turnovers of lauric acid were observed over the 3 cycles?

Response: We agree with the review that there is a decrease in performance of PdNP/H-MZSM5, which is especially clear for the 2nd recycle, and thus, we have amended the manuscript to be clearer. The TON for the three recycles are 688, 625 and 574.

Action: Manuscript amended

Catalyst recycling studies show that PdNP/H-MZSM5-DA exhibits good recyclability (Supplementary Fig. 15 and 16) without requiring reactivation between runs, whilst PdNP/H-MZSM5 shows reasonable recyclability (TON drop from 688 to 574 over 3 cycles).

4. The carbon balance of ~80% in the optimization studies is concerning, especially as the authors have shown that coking does not occur in the system. The reason or source of this carbon imbalance needs to be identified.

Response: As identified by TGA analysis of the spent catalyst, some of the missing mass balance is from substrate, intermediate and product adsorption. As per reviewer three comment, low solubility of the substrate is also an issue, and while sampling hot aims to overcome this, some substrate is likely lost. Given our assessment of performance, including benchmarking against other systems, includes looking at the products, which are soluble at room temperature, the impact of this loss is minimal and affects conversion solely.

Action: Manuscript amended

So, while species are adsorbed, which accounts for some of the mass balance of the process (the remaining missing mass may reflect low solubility and precipitation of the substrate during sampling), it is reasonable to assume they remain as species that partake in the cascade process.

It is important to note that due to the potential for precipitation of the fatty acid substrate at room temperature, sampling should be conducted at reaction temperature, and the sampling system should be flushed prior to sample collection to avoid carryover. Loss of the substrate during sampling will impact solely on the conversion reported. There are no issues with regard to product solubility.

5. The authors claim that the species adsorbed onto the zeolite participate in the cascade process. There is no data to support this. This can be done by separating the spent zeolites and running a reaction without any additional reactant.

Response: Depending on the nature of adsorbed species, it may be reasonable to assume they could participate. TGA analysis of the spent catalysts reveals the majority of mass loss occurs from 200 to 300 °C, which corresponds with the boiling points of dodecene and dodecane (213 and 216 °C) and dodecanol, dodecanal and dodecanoic acid (259, 242, 299 °C). Thus, we have inferred that the adsorbed species are these compounds, rather than coke, which would desorb at higher temperatures (and would not be able to participate in the cascade). We have edited the text to clarify this assumption in the nature of the adsorbed species. The suggested experiment is logical; however, the reduced mass of recovered catalysts would result in very low concentrations of substrate/products (towards the lower detection limits) and, hence, greater uncertainty of the results.

Action: Manuscript amended

Thermogravimetric analysis of PdNP/H-MZSM5-DA (Supplementary Fig. 17) shows mass losses only below 300 °C, which we attributed to arise from the desorption of adsorbed substrate (dodecanoic acid b.p. 299 °C), intermediates (dodecanal b.p. 242 °C, dodecanol b.p. 259 °C, and dodecene b.p. 213 °C), and product (dodecane b.p. 216 °C), whereas combustion

of coke species requires temperatures ≥ 500 °C. So, while species are adsorbed, which accounts for some of the mass balance of the process (the remaining missing mass may reflect low solubility and precipitation of the substrate during sampling), it is reasonable to assume they remain as species that partake in the cascade process.

6. Fatty acids such as lauric are derived in a range of carbon chain lengths (C8-C18). Can the authors demonstrate the activity of the zeolite for other commonly found fatty acids?

Response: The review makes a fair comment, and given the importance of expanding this study to demonstrate the advancement realised from this work and the application of this catalyst, we have expanded the study to include palmitic (C16) and stearic (C18) acids, both common compounds found in plant oils.

Action: Manuscript and ESI amended.

The inherent benefit of the unique active site spatial compartmentalisation within PdNP/H-MZSM5-DA is further evident from its capacity for HDO of other fatty acids, including palmitic (C16) and stearic (C18) (Supplementary Table 5).

Supplementary Table 5. PdNP/H-MZSM5-DA performance for HDO of fatty acids.^a

Substrate	Acid Conv., %	C _n Sel., %	C _{n-1} Sel., %	Production rate ^b
Palmitic acid	19.2	52.4	14	13.6
Stearic acid	17.5	78.4	21.6	12.5

^a Reaction conditions: 34 mg 1 wt.% Pd-doped catalyst, 380 mg palmitic or 420 mg stearic acid, 0.1 cm³ of nonane as the internal standard, 40 cm³ of hexane as the solvent, at 200 °C, 30 bar and 1000 rpm for 6 h. ^b Production rate was calculated based on the yield of C_n in 2 h. The errors of conversion and selectivity are within $\pm 3\%$, whereas yields and production rates are within $\pm 5\%$.

7. Fatty acids can be relatively expensive to source compared with aviation fuels; recent work has shown that they can be upgraded to higher-value surfactants and lubricants through ketonization chemistry. Can the authors comment on the economics of this process?

Response: While the reviewer is correct to highlight the cost of fatty acids (and triglycerides), we do point out that the techno-economic analysis of sustainable aviation fuels (SAF) from these feedstocks is favourable compared to other biomass feeds (Biomass and Bioenergy, 2021, 145, 105942). While the biggest cost factor is the feedstock development and advancement in catalytic processes, including ones that can operate at low temperatures such as the system reported here, will decrease process economics and increase the favourability of such production processes. While a detailed economic analysis of the process is beyond the scope here, we have amended the introduction to highlight the potential of SAF from fatty acids (and triglycerides) with the appropriate reference.

Action: Manuscript amended

Hydrodeoxygenation (HDO) of fatty acids (and triglycerides) to alkanes, ideally under mild reaction conditions (< 300 °C), provides one of the most promising economically viable solutions being investigated^{29,30}.

Minor Comments

1. In Table.1 The authors need to report the turnover numbers for each catalyst screened

Response: TONs have been added to Table 1.

Action: Manuscript amended

Table 1 | Lauric acid HDO performance over conventional prepared and spatial compartmentalised catalysts.^a

Catalysts	Acid Conv., %	C ₁₂ Sel., %	C ₁₁ Sel., %	Productivity ^b (mol _{C₁₂} mol _{Pd} ⁻¹ h ⁻¹)	TON ^c
H-MZSM5	0	0	0	0	0
PdNP/Na-MZSM5	2.2	27.3 (alcohol)	0 ^d	0.05 (alcohol)	29
H-MZSM5 + PdNP/Na-MZSM5 ^e	20.6	28.1	2.1	1.7	278
PdNP/H-MZSM5	51.0	51.8	9.9	11.9	689
Pd _{imp} /H-MZSM5	31.2	38.4	20.5	3.6	187
PdNP/ H-MZSM5-DA	94.1	64.9	12.6	30.6	1272
Pd _{imp} / H-MZSM5-DA	70.9	39.9	18.2	11.3	426
PdNP _{small} /H- MZSM5	50.9	40.4	14.9	9.9	688
PdNP _{small} /H- MZSM5-DA ^f	71.5	59.3	24.4	28.1	1205

^a Reaction conditions: 100 mg 1 wt.% Pd-doped catalyst, 300 mg lauric acid, 0.1 cm³ of nonane as the internal standard, 40 cm³ of hexane as the solvent, at 200 °C, 30 bar and 1000 rpm for 6 h. ^b Production rate was calculated based on the yield of n-C₁₂ + iso-C₁₂ in 2 hours, as shown in Fig. 4 (a). The unit is mol_{dodecane} mol_{Pd}⁻¹ h⁻¹, in which the quantity of Pd is based on Pd loading from ICP. ^c TON calculated based on moles of lauric acid converted at 6 h by surface Pd. ^d The concentration of C₁₁ is below the GC detection limit. ^e Physical mixture of 100 mg H-MZSM5 and 100 mg PdNP/Na-MZSM5. ^f 80 mg 1 wt.% Pd-doped catalyst. The errors of conversion and selectivity are within ±3%, whereas yields and production rates are within ±5%.

2. The reaction conditions i.e. temperature reactant loading need to be reported in figure 4 and the other supplementary figures where yield data is given.

Response: Reaction conditions have been added where they were missing.

Action: Manuscript and ESI amended

Fig. 4 | HDO of lauric acid over different Pd catalysts. Reaction conditions: 100 mg 1 wt.% Pd-doped catalyst, 300 mg lauric acid, 0.1 cm³ of nonane as the internal standard, 40 cm³ of hexane as the solvent, at 200 °C, 30 bar and 1000 rpm for 6 h.

Supplementary Figure 11. Process conditions optimisation for lauric acid HDO over PdNP/H-MZSM5.

Reaction profiles as a function of temperature (a) 180 °C, (b) 200 °C, (c) 220 °C; (d) product distribution of the reaction systems after the 6-h reaction under different pressures; (e) reaction profile as a function of stirring rate. The errors of conversion are 3% and yields are ±5%. Reaction conditions: 100 mg 1 wt.% Pd-doped catalyst, 300 mg lauric acid, 0.1 cm³ of nonane as the internal standard, 40 cm³ of hexane as the solvent, unless stated otherwise at 200 °C, 30 bar H₂ and 1000 rpm.

Supplementary Figure 14. Lauric acid reaction profiles for PdNP/H-MZSM5-DA, PdNP/H-MZSM5, PdNP_{small}/H-MZSM5, PdNP_{imp}/H-MZSM5, and physical mixture of H-MZSM5 + PdNP/Na-MZSM5. Error bars correspond to the standard deviation of the average taken over at least two independent measurements. Reaction conditions: 100 mg 1 wt.% Pd-doped catalyst, 300 mg lauric acid, 0.1 cm³ of nonane as the internal standard, 40 cm³ of hexane as the solvent, at 200 °C, 30 bar and 1000 rpm.

Supplementary Figure 15. Stability test for HDO of lauric acid. (a) Product yields from recycle studies for lauric acid HDO over (a) PdNP/H-MZSM5 and (b) PdNP/H-ZSM5. (c) Representative HAADF-STEM image of PdNP/H-MZSM5 after the second recycle (with EDS mapping). (d) Representative HAADF-STEM image of PdNP/H-ZSM5 after the second recycle (with EDS mapping). (e) Representative TEM image of the fresh PdNP/H-MZSM5; (f) Representative TEM image of the fresh PdNP/H-ZSM5. Error bars in (a) and (b) correspond to the standard deviation of the average taken over at least two independent measurements. Reaction conditions: 100 mg 1 wt.% Pd-doped catalyst, 300 mg lauric acid, 0.1 cm³ of nonane as the internal standard, 40 cm³ of hexane as the solvent, at 200 °C, 30 bar and 1000 rpm.

Supplementary Figure 16. (a) Product yields from recycle studies for lauric acid HDO over PdNP/H-MZSM5-DA. (b) Representative HAADF-STEM image of PdNP/H-MZSM5-DA after the second recycle (with EDX mapping). The errors of yields are within ±3%. Reaction conditions: 100 mg 1 wt.% Pd-doped catalyst, 300 mg lauric acid, 0.1 cm³ of nonane as the internal standard, 40 cm³ of hexane as the solvent, at 200 °C, 30 bar and 1000 rpm.

3. Does the system have any external mass transfer limitations? Have the authors looked into testing the production rate at different stirring speeds?

Response: The impact of agitation rate was studied for PdNP/H-MZSM5, as shown in Supplementary Figure 11 (e). Decreasing the stirring rate showed no impact on conversion or productivity. Therefore, we conclude that external mass diffusion is not a limiting factor at 1000 RPM.

Action: Manuscript amended

The impact of stirring rate shows no effect, as presented in Supplementary Fig. 11, ruling out bulk mass diffusion effects.

REVIEWERS' COMMENTS

Reviewer #2 (Remarks to the Author):

The authors have addressed my concerns, now I suggest its publication in Nature Communications.

Reviewer #3 (Remarks to the Author):

Comments have mostly been addressed, improving this interesting manuscript, only aspects noted could have been strengthened slightly more.

Reviewer #4 (Remarks to the Author):

The author has addressed all my comments, i recommend the article for publication